Analysis

# Predicting visual function by interpreting a neuronal wiring diagram

H. Sebastian Seung[1✉]

As connectomics advances, it will become commonplace to know far more about the structure of a nervous system than about its function. The starting point for many investigations will become neuronal wiring diagrams, which will be interpreted to make theoretical predictions about function. Here I demonstrate this emerging approach with the *Drosophila* optic lobe, analysing its structure to predict that three Dm3 (refs. 1–4) and three TmY (refs. 2,4) cell types are part of a circuit that serves the function of form vision. Receptive fields are predicted from connectivity, and suggest that the cell types encode the local orientation of a visual stimulus. Extraclassical[5,6] receptive fields are also predicted, with implications for robust orientation tuning[7], position invariance[8,9] and completion of noisy or illusory contours[10,11]. The TmY types synapse onto neurons that project from the optic lobe to the central brain[12,13], which are conjectured to compute conjunctions and disjunctions of oriented features. My predictions can be tested through neurophysiology, which would constrain the parameters and biophysical mechanisms in neural network models of fly vision[14].

The 40,000 neurons in an optic lobe of the *Drosophila* brain[15–17] have now been completely classified into cell types[4]. Although the 200+ cell types intrinsic to the optic lobe are well defined by structural analysis, their functions are largely unknown because less than 20% have had their visual responses recorded by neurophysiologists. For the most part, at present the online catalogue of visual cell types (https://codex.flywire.ai/app/optic_lobe_catalog) is an enigmatic museum of structures without functions.

A beautiful example is Dm3, a trio of neuronal cell types that are intrinsic to the distal medulla. A Dm3 cell is tangentially oriented (Fig. 1a), and is orthogonal to the columnar cells that project from the medulla to other neuropils (Fig. 1a). The three Dm3 types are easily distinguished from each other because their dendrites point in three directions[4] (Fig. 1b–d and Methods). For interactive visualizations of cells and pathways, see the URLs in Supplementary Data 1.

Dm3 was originally called the line amacrine cell when it was discovered in 1970 in dipterans[1]. It was named Dm3 in *Drosophila* in 1989[2], and split into two types in 2015[3,18,19]. A third Dm3 type was recently identified in a large-scale connectomic census[4]. No recordings of Dm3 visual responses have ever been reported by physiologists. Dm3 function has remained unknown.

I will begin by attempting to predict Dm3 function from structure. The field of connectomics has been motivated by a conviction that the computational capabilities of the brain depend strongly on the connectome[20,21]. Indeed, neural circuit functions such as the computation of visual motion[22], heading direction[23], and reward and punishment signals[24] are now known to be supported by specific patterns of neural connectivity in the *Drosophila* brain. These successes of structural explanation came after neurophysiologists had already identified circuit functions[25–27]. The present study attempts to go beyond explaining post facto. I tackle the challenge of predicting function ex ante, a more stringent test of the power of structure–function relationships.

## Predictions of Dm3 receptive fields

The receptive field of a visual neuron was classically defined as "the region of the retina which must be illuminated in order to obtain a response"[28]. I predict Dm3 receptive fields by mapping their input connections from Tm1, a cell type that is in one-to-one correspondence with the hexels (hexagonal pixels) of the compound eye (Methods). Tm1 is the strongest input to Dm3 (Extended Data Fig. 1a,c), and Dm3 is a strong output of Tm1 (Extended Data Fig. 2d). Figure 1e shows a typical Dm3 cell receiving strong Tm1 input in three collinear columns, even though the cell extends over many columns (Fig. 1b–d).

The Tm1 cells presynaptic to each Dm3v cell were mapped to a hexagonal lattice (Fig. 1f, Methods and Supplementary Data 2). The colour of each hexel indicates the number of synapses received by the Dm3v cell from Tm1 cells. Aligning (Fig. 1g) and averaging all maps yielded an average Tm1–Dm3v input map (Fig. 1b inset).

This procedure was repeated for all three Dm3 types. The resulting average maps (Fig. 1b–d insets) show that the three Dm3 types are aligned to the three cardinal orientations of the hexagonal lattice[29] (*p*, *q* and *v* axes in Fig. 1f).

Cell-to-cell variation is visible in the individual maps for single Dm3 cells (Fig. 1g and Supplementary Data 3). To quantify variability, each Tm1–Dm3 connectivity map was approximated as an ellipse (Fig. 1g,h and Methods). The distributions of angular orientations for the Dm3 populations overlap little (Fig. 1i), and the median angles are close to the cardinal orientations of the hexagonal lattice. The aspect ratios of the ellipses have median values near 4 for all Dm3 populations (Fig. 1j).

To further characterize the size and shape of the predicted receptive fields, one-dimensional (1D) projections of Tm1–Dm3 maps were computed (Fig. 1k–m). The longitudinal and transverse projections provide information about the length and width of the predicted receptive field,

[1]Neuroscience Institute and Computer Science Department, Princeton University, Princeton, NJ, USA. ✉e-mail: sseung@princeton.edu

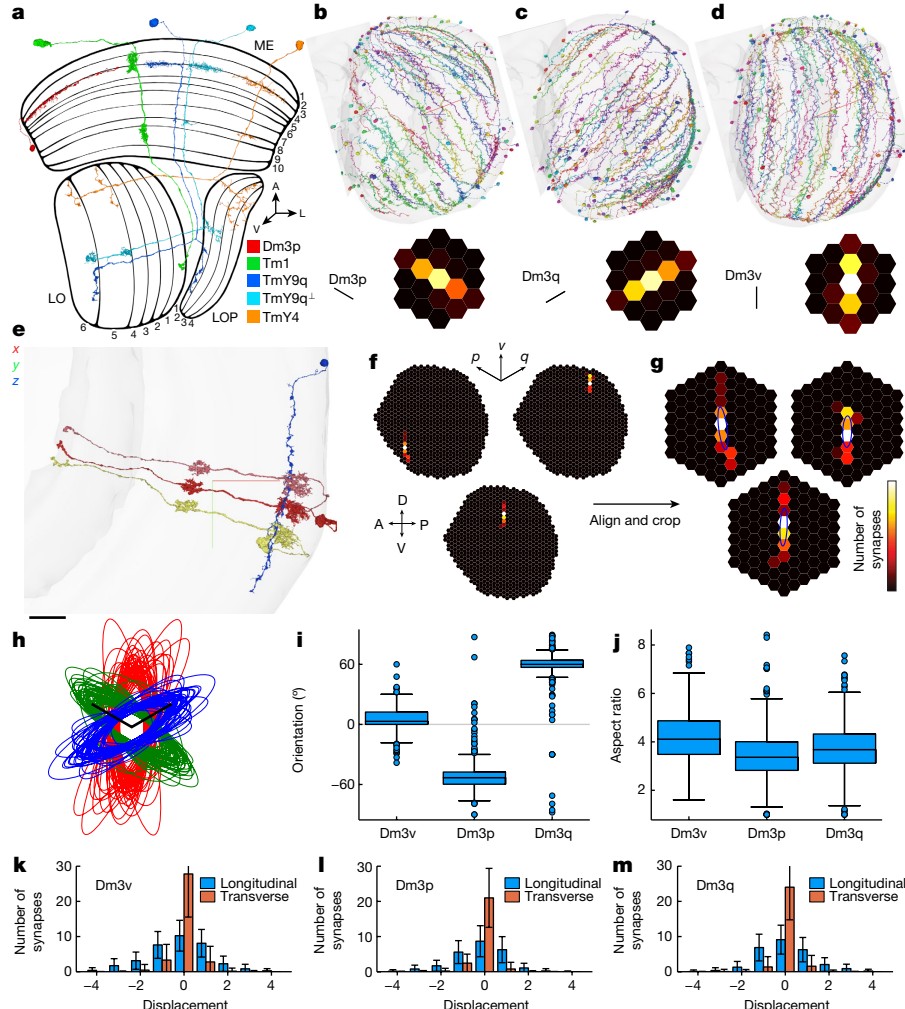

**Fig. 1 | Dm3 receptive fields predicted by mapping presynaptic Tm1 cells.**
**a**, A Dm3 (red) interneuron local to the distal medulla (ME). Tm1 (green) projects from the medulla to the lobula (LO). TmY9 (blue and cyan) and TmY4 (orange) project from the medulla to the lobula and the lobula plate (LOP). ME, LO, and LOP have 10, 6, and 4 layers (numerical sequences), respectively[2]. **b–d**, Dm3 types with predicted receptive fields. Dm3p, Dm3q and Dm3v point in the posteroventral (**b**), posterodorsal (**c**) and ventral (**d**) directions. Insets: the average receptive field predictions are oriented along the cardinal directions of the hexagonal lattice. **e**, A Dm3v cell (blue) with all strong (>4 synapses) presynaptic partners, here all Tm1. Scale bar, 15 μm. **f,g**, Computation of average receptive field predictions. Presynaptic Tm1 cells were mapped to predict receptive fields of individual cells (Dm3v examples in **f**). Receptive field predictions were aligned, cropped and averaged to yield the insets of **b–d**. The heat map represents the synapse number. **h**, Aligned ellipse approximations to predicted receptive fields of individual Dm3v (red), Dm3p (green) and Dm3q

(blue) cells (see also examples in **g**). Scale bars, one lattice constant (*p* and *q* axes). **i**, Ellipse orientations. Positive angle is clockwise relative to vertical (0°). **j**, Ellipse aspect ratios. **k–m**, 1D projections of average receptive field predictions onto the longitudinal and transverse axes: vertical, horizontal for Dm3v (**k**); *p, p*⊥ for Dm3p (**l**); *q, q*⊥ for Dm3q (**m**). The unit of longitudinal (transverse) displacement is lattice constant (×$\sqrt{3}$/2). Error bars indicate standard deviation across Dm3 cells. Crosshairs (**b–e**) are lateral (red), ventral (green) and posterior (blue). Scale bar in orthographic projection (**e**), 15 μm. Heat map maxima (white): 9 (**b–d**); 14, 7 and 10 synapses (**f,g**, clockwise from bottom). Statistics over 498 Dm3p, 454 Dm3q and 321 Dm3v cells, represented by random subsets of 50 cells in **b–d,h**. Maps are based on 745 Tm1 cells. The whisker–box distance is 1.5× the interquartile range (box), with only data points outside whiskers shown (**i,j**). For interactive visualizations of the cells in **b–e**, see Supplementary Data 1. A, anterior; D, dorsal; L, lateral; V, ventral; P, posterior.

respectively. Standard deviations are marked to provide information about variability across cells.

The Tm1 receptive field is radially symmetric, with a centre that is about one ommatidium wide[30]. As this is so narrow, Tm1–Dm3 connectivity maps (Fig. 1b–d insets) can be regarded as estimates of Dm3 receptive fields, assuming that a Dm3 cell sums inputs from its presynaptic Tm1 cells.

Orientation selectivity was discovered in mammalian primary visual cortex, and found to be related to spatial orientation of the receptive field[31]. As the predicted receptive fields of Dm3 are oriented, I likewise predict that Dm3 cells prefer stimuli at the three cardinal orientations.

Although Tm1 is the strongest input to Dm3 by a large margin (Extended Data Fig. 1a), Dm3 also receives weaker input from other

'hexel cell types' (see Methods for definition). Maps of their connectivity with Dm3 look similar to Tm1–Dm3 maps (Extended Data Fig. 3). Like Tm1, these inputs (Mi4, Tm2, L3 and Tm9) are known to have receptive field centres that are roughly a single ommatidium in width[30,32,33], and therefore are not expected to change the shape of the receptive field estimated from Tm1. All of these inputs are consistent with the prediction that Dm3 cells have OFF receptive fields (Methods).

## Predictions of TmY receptive fields

Top targets of Dm3 include TmY4 and TmY9 (Extended Data Fig. 1b,d), which project from the distal medulla to the lobula and lobula plate[2] (Fig. 1a). In the distal medulla, TmY4 dendrites are horizontally

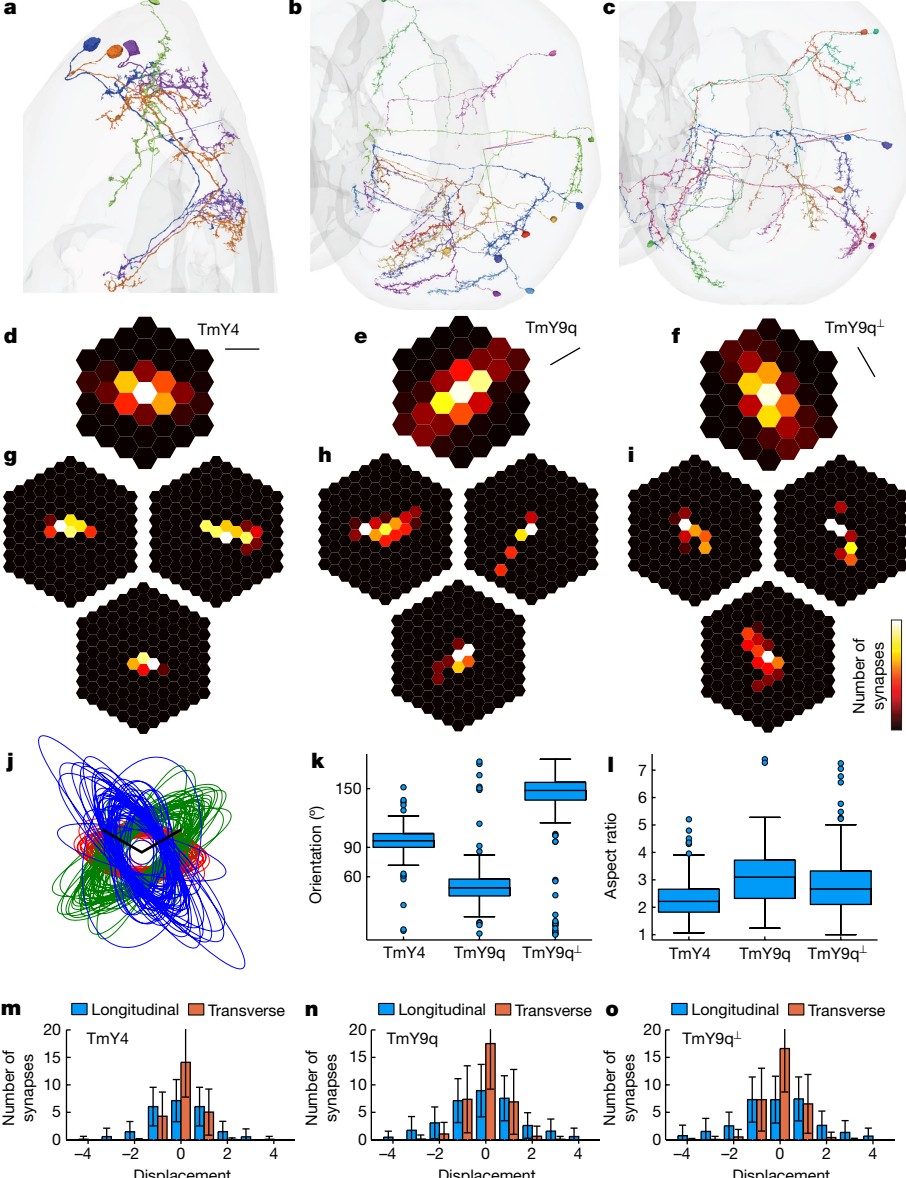

**Fig. 2 | TmY receptive fields predicted by mapping presynaptic Tm1 cells.**
**a**, TmY4 cells presynaptic to a Dm3v cell (green). TmY4 dendrites in the distal medulla are horizontally oriented and typically extend symmetrically on either side of the main trunk. **b**, TmY9q dendrites in the distal medulla point in the anteroventral direction, roughly antiparallel to Dm3q dendrites. **c**, TmY9q$^\perp$ dendrites in the distal medulla point in the posteroventral direction, roughly orthogonal to Dm3q and TmY9q dendrites. **d**–**f**, Predicted receptive field, averaged over cells, for TmY4 (**d**), TmY9q (**e**) and TmY9q$^\perp$ (**f**). **g**–**i**, Receptive fields of individual TmY4 (**g**), TmY9q (**h**) and TmY9q$^\perp$ (**i**) cells predicted by mapping presynaptic Tm1 cells, after aligning and cropping. The heat map represents the synapse number. **j**, Aligned ellipse approximations to predicted receptive fields of individual TmY4 (red), TmY9q (green) and TmY9q$^\perp$ (blue) cells.

Scale bars, one lattice constant ($p$ and $q$ axes). **k**, Ellipse orientations. Positive angle is clockwise relative to vertical (0°). **l**, Ellipse aspect ratios. **m**–**o**, 1D projections of average receptive field predictions onto the longitudinal and transverse axes: horizontal, vertical for TmY4 (**m**); $q$, $q^\perp$ for TmY9q (**n**); $q^\perp$, $q$ for TmY9q$^\perp$ (**o**). The unit of longitudinal (transverse) displacement is the lattice constant ($\times\sqrt{3}/2$) for **n**, and vice versa for **m**,**o**; that is, longitudinal and transverse units are swapped. Error bars indicate standard deviation across TmY cells. Crosshairs in **a**–**c** are lateral (red), ventral (green) and blue (posterior). Heat map maxima (white) are 5.3, 4.8 and 4.8 synapses clockwise from bottom (**d**–**f**). Statistics over 211 TmY4, 172 TmY9q and 189 TmY9q$^\perp$ cells, with random subsets of 50 cells shown in **j**. Box plot conventions are as in Fig. 1. For interactive visualizations of cells, see Supplementary Data 1.

oriented, extending symmetrically on two sides of the main trunk (Fig. 2a). TmY9 has been split into TmY9q and TmY9q$^\perp$ (Methods). In the distal medulla, both TmY9 types have dendrites that are asymmetrically directed to one side of the main trunk along the $q$ and $q^\perp$ directions.

Tm1 is one of the top inputs to TmY4 and TmY9 (Extended Data Figs. 4 and 5). Once again, I predict receptive fields by mapping Tm1 inputs. The average maps have three distinct orientations (Fig. 2d–f). Cell-to-cell variability (Fig. 2g–i and Supplementary Data 4) is visualized

by approximating ellipses (Fig. 2j). The median orientations of the TmY4 and TmY9q$^\perp$ ellipses are well approximated by the horizontal and $q^\perp$ orientations of the hexagonal lattice (Fig. 2k). The median orientation of the TmY9q ellipses is between the $q$ and $p^\perp$ orientations, and closer to $q$ (Fig. 2k). Roughly speaking, each TmY orientation is orthogonal to one of the Dm3 orientations, although the orthogonality of TmY9q and Dm3p is less than perfect. Aspect ratios of the ellipses indicate the degree of anisotropy (Fig. 2l). Longitudinal and transverse 1D projections of Tm1–TmY maps (Fig. 2m–o) provide complementary

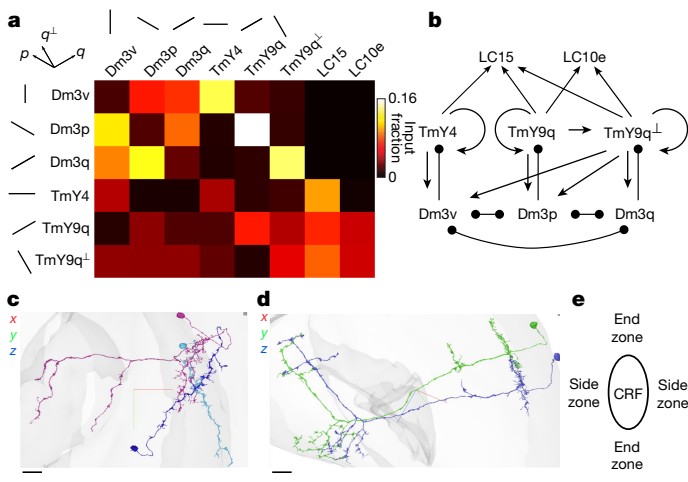

**Fig. 3 | Connectivity between Dm3, TmY and LC cell types. a**, A type-to-type connection matrix as a heat map indicating the fraction of input synapses to a postsynaptic type (column) originating from a presynaptic type (row). Next to each type name, a line segment indicates the dendritic orientation in the distal medulla. **b**, A type-to-type wiring diagram of connections in **a** that represent at least 3% of the input synapses to the postsynaptic target type. Lines ending in arrowheads (circles) indicate connections that are presumed excitatory (inhibitory). Dm3 cells prefer to synapse onto Dm3 and TmY cells with orthogonal or roughly orthogonal dendrites. TmY cells mostly prefer to synapse onto TmY cells of the same type, but TmY9q also synapses onto TmY9q$^{\perp}$. TmY4 and TmY9q prefer to synapse onto Dm3 cells with orthogonal dendrites, but TmY9q$^{\perp}$ synapses onto all Dm3 types. LC15 receives input from all TmY types, whereas LC10e receives input from TmY9q and TmY9q$^{\perp}$. **c**, The Dm3q–Dm3p–TmY9q pathway. Dm3q (blue) disinhibits TmY9q (purple) through the intermediary Dm3p (cyan), assuming that Dm3 is inhibitory. Dm3p also synapses onto Dm3q, so they are reciprocally connected. **d**, A pair of TmY4 cells with strong (>4 synapses) reciprocal connections. **e**, Physiologists have shown that the neural response to a stimulus presented inside the CRF may be modulated by a stimulus presented outside the CRF[5,6]. The size and sign of the effect often depends on whether the modulating stimulus is located in the end zones versus the side zones of an oriented CRF. The effect may also depend on other properties of the modulating stimulus, such as its orientation or contrast. Scale bars in orthographic projections (**c,d**), 10 μm. Crosshairs are lateral (red), ventral (green) and posterior (blue). For interactive visualizations of the pathway in **c** and the cells in **d**, see Supplementary Data 1.

information about the length and width of the predicted receptive field, respectively.

From these estimates of receptive fields, TmY4, TmY9q and TmY9q$^{\perp}$ are predicted to prefer visual stimuli at the horizontal, $q$ and $q^{\perp}$ orientations, respectively. On the basis of their strong Tm1 inputs, TmY4 and TmY9 are predicted to be OFF cells, although they also receive input from the ON hexel type Mi1 (Extended Data Figs. 4 and 5). Monosynaptic inputs to individual TmY cells from Tm1, Mi1 and other hexel types are mapped in Supplementary Data 4.

## Connectivity between Dm3 and TmY types

Dm3 and TmY neurons are also connected with each other (Fig. 3a and Extended Data Figs. 1, 4 and 5). Dm3 will be presumed inhibitory in its synaptic effects on other cells, and TmY will be presumed excitatory, on the basis of evidence from both electron micrographs and transcriptomic data (Methods).

Each Dm3 type avoids synapsing onto cells of the same type, and prefers to synapse onto the other two Dm3 types, which have different dendrite orientations (Fig. 3a). Each Dm3 type prefers to synapse onto a single TmY type, that with the orthogonal dendrite orientation (Fig. 3a). Therefore, Dm3 output connectivity conforms to cross-orientation

inhibition, a connectivity motif proposed for visual cortex more than 50 years ago[34,35].

TmY cells prefer to synapse onto TmY cells of the same type (Fig. 3a). It follows that TmY–TmY connectivity conforms approximately to iso-orientation excitation, a connectivity motif proposed for visual cortex more than 40 years ago[8]. In addition, there is some weaker connectivity from TmY9q to TmY9q$^{\perp}$ (Fig. 3a).

TmY4 and TmY9q cells synapse onto Dm3 cells of the orthogonal orientation (Fig. 3a). By contrast, TmY9q$^{\perp}$ shows no preference, synapsing onto Dm3 cells of all orientations (Fig. 3a). Therefore TmY-to-Dm3 connectivity is a combination of cross-orientation and indiscriminate connectivity.

The above connectivity patterns are represented schematically in Fig. 3b. This is effectively a thresholded version of Fig. 3a that will be used in subsequent analyses.

Figure 3c illustrates a Dm3 cell synapsing onto another Dm3 cell, which in turn synapses onto a TmY cell. Figure 3d illustrates reciprocally connected TmY4 cells. These visualizations suggest that there are spatial relationships (Extended Data Fig. 6) between the connected cells, which are not described by population-level wiring diagrams (Fig. 3a,b). The consequences of these spatial relationships for visual responses of cells will be predicted below.

## Dm3 extraclassical receptive fields

The receptive field predictions of Figs. 1 and 2 were for the 'classical' receptive field (CRF). For some visual neurons, stimulation of areas outside the CRF may modulate the response to stimulation of the CRF[5]. These areas are sometimes called the 'extraclassical' receptive field (ERF). By definition, stimulating the ERF alone (without the CRF), should not result in any response. The amount of modulation induced by ERF stimulation may depend on the orientation, contrast or other properties of the modulating stimulus[6]. Locations outside an oriented CRF near the longitudinal axis will be called the end zones, and locations near the transverse axis will be called the side zones (Fig. 3e).

I propose that the ERF of Dm3 cells is determined by disynaptic pathways from Tm1 to Dm3. (This is a conceptual leap, and potential pitfalls are deferred to the section entitled 'Predictions and their limitations'). I searched for such pathways with the greatest anatomical strength. In brief, the strength of the A-to-B connection was defined as the fraction of input synapses to cell type B provided by cell type A. The strength of the A-to-B-to-C disynaptic pathway was then defined as the product of the A-to-B and B-to-C strengths. Details about this strength estimate and its limitations are given in the Methods.

Strong disynaptic pathways from Tm1 to Dm3 were mapped (Methods, Extended Data Fig. 7 and Supplementary Data 3), aligned and averaged across Dm3 cells (Fig. 4a–c). For each Dm3 type, the average disynaptic pathway map overlaps with the average predicted CRF (Fig. 1b,c), but also extends beyond it. The beyond-the-CRF portion of each disynaptic pathway map will be regarded as a component of the predicted ERF. Henceforth, 'pERF' and 'pCRF' will be used as abbreviations for the predicted ERF component and predicted CRF, respectively. As the maps do not have sharp borders, they will be approximated as ellipses for the purpose of defining beyond-the-CRF regions (Fig. 4a–c).

For each Dm3 type, the strongest two pERFs are from the other two Dm3 types. These mostly extend into the side zones of the pCRF (Fig. 4a–c), and are predicted to be suppressive because Dm3 is presumed inhibitory. Such cross-orientation suppression could have the function of sharpening orientation tuning[7].

The next strongest pERF is mediated by T2a, and is predicted to be facilitating because T2a is presumed excitatory (Methods). The T2a-mediated average pERF covers both end zones of the Dm3v pCRF (Fig. 4c), and mainly the posterior end zone of the Dm3p and Dm3q pCRFs (Fig. 4a,b). Ellipse approximations to Tm1–T2a–Dm3 maps for individual Dm3 cells show that their orientations tend to be similar

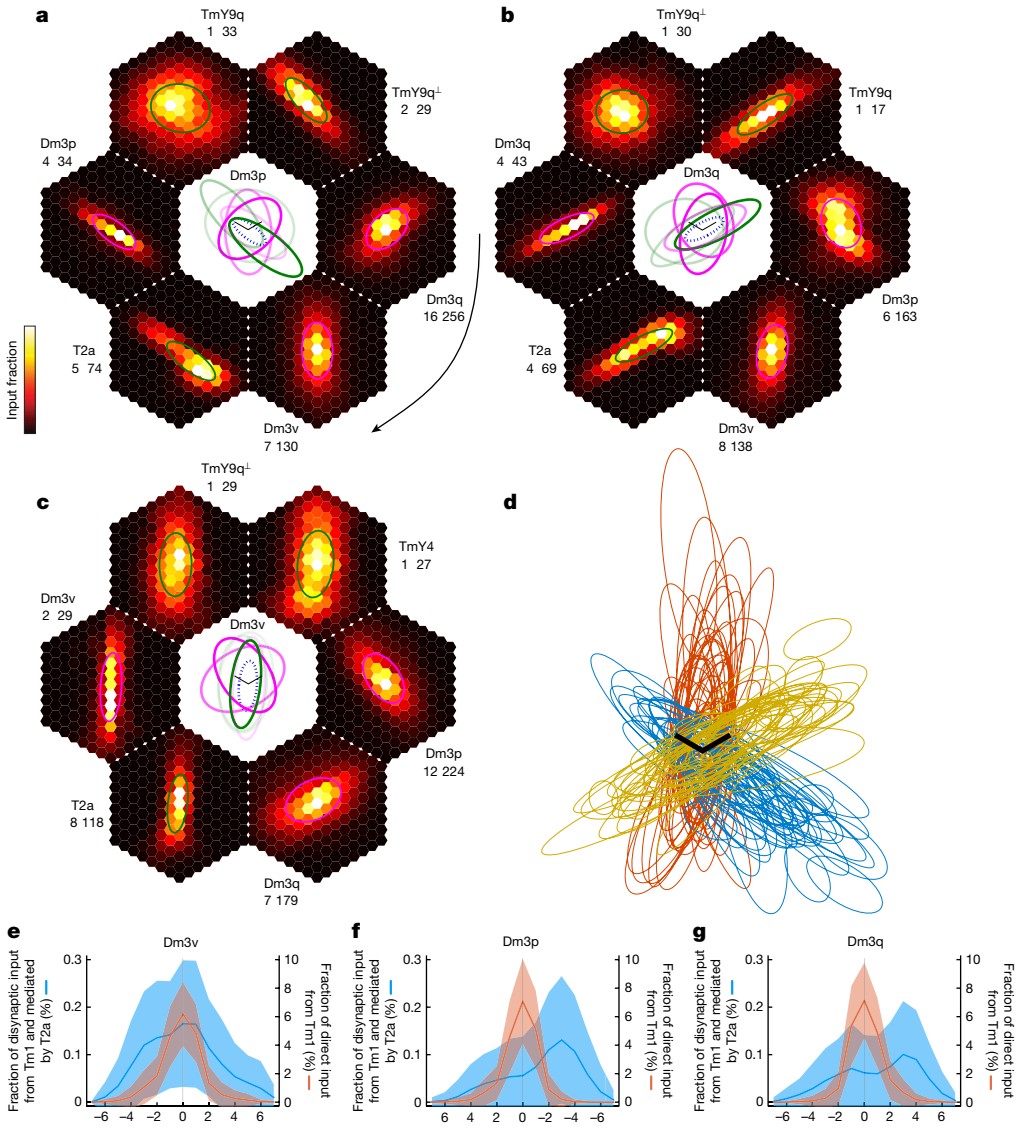

**Fig. 4 | Dm3 ERFs predicted by mapping disynaptic pathways. a–c,** Heat maps (hexagons) of disynaptic pathways from Tm1 to Dm3, aligned and averaged over Dm3p (**a**), Dm3q (**b**) and Dm3v (**c**) cells. The portion of a disynaptic map outside the pCRF is a predicted ERF component mediated by the named intermediary type (text next to map). Pathways are ranked clockwise (curved arrow) by anatomical strength (Extended Data Fig. 7, green lines). The numbers indicate the fraction of disynaptic input in units of 0.01%: heat map maximum (white) and spatial sum over hexels. For each Dm3 type, the top two pathways are mediated by the two other Dm3 types, and the third strongest pathway is mediated by T2a. Ellipse approximations are superimposed on each map, with magenta (green) signifying pathways presumed inhibitory (excitatory). Centre: ellipses are shown again, anchored on the average pCRF (dotted ellipse) and enlarged for visibility by ×1.5. Ellipse opacity represents anatomical strength relative to the strongest inhibitory (magenta) or excitatory (green) pathway. **d,** Aligned ellipse approximations to Tm1–T2a–Dm3 maps for 50 representative Dm3p (blue), Dm3q (gold) and Dm3v (brown) cells. **e–g,** T2a-mediated disynaptic maps from Tm1 to Dm3 extend beyond the pCRFs, as shown by 1D longitudinal projections of average maps. A pERF component is predicted in both pCRF end zones for Dm3v (**e**), and mainly in the posterior end zone for Dm3p (**f**) and Dm3q (**g**). The unit of displacement is lattice constant. Ribbons represent standard deviation across Dm3 cells. Dm3 and Tm1 sample sizes are as in Fig. 1, and 866 T2a cells were included. Scale bars (**a–c** centre, **d**), one lattice constant (*p* and *q* axes).

to those of the Dm3 pCRFs (Fig. 4d). For some Dm3p and Dm3q cells, the Tm1–T2a–Dm3 map is small and located in the pCRF end zone, not overlapping with the pCRF at all. This shows up as a hotspot in the average Tm1–T2a–Dm3 maps (Fig. 4a,b).

1D projections of the average Tm1–T2a–Dm3 maps (Fig. 4e–g) show that they tend to be longer than the pCRFs, and result in pERFs that are biased towards the posterior pCRF end zone of Dm3p (Fig. 4f) and Dm3q (Fig. 4g).

The remaining disynaptic pathways from Tm1 to Dm3 are substantially weaker by anatomical strength (Extended Data Fig. 7), but are included for completeness because they might turn out to be physiologically strong. It is not easy to predict the overall effect at locations where suppressive and facilitating pERFs overlap. I predict that suppression will be prominent in the pCRF side zones, where the Dm3-mediated pERFs seem numerically stronger. In the end zones, the T2a-mediated pERFs are largely unopposed by suppression, and should lead to facilitation that is not selective for orientation or contrast of the modulating stimulus.

pERFs for individual Dm3 cells are provided in Supplementary Data 3. There is considerable variability across cells of the same type, but the geometric relationships evident in the average pERFs do hold for many individual cells.

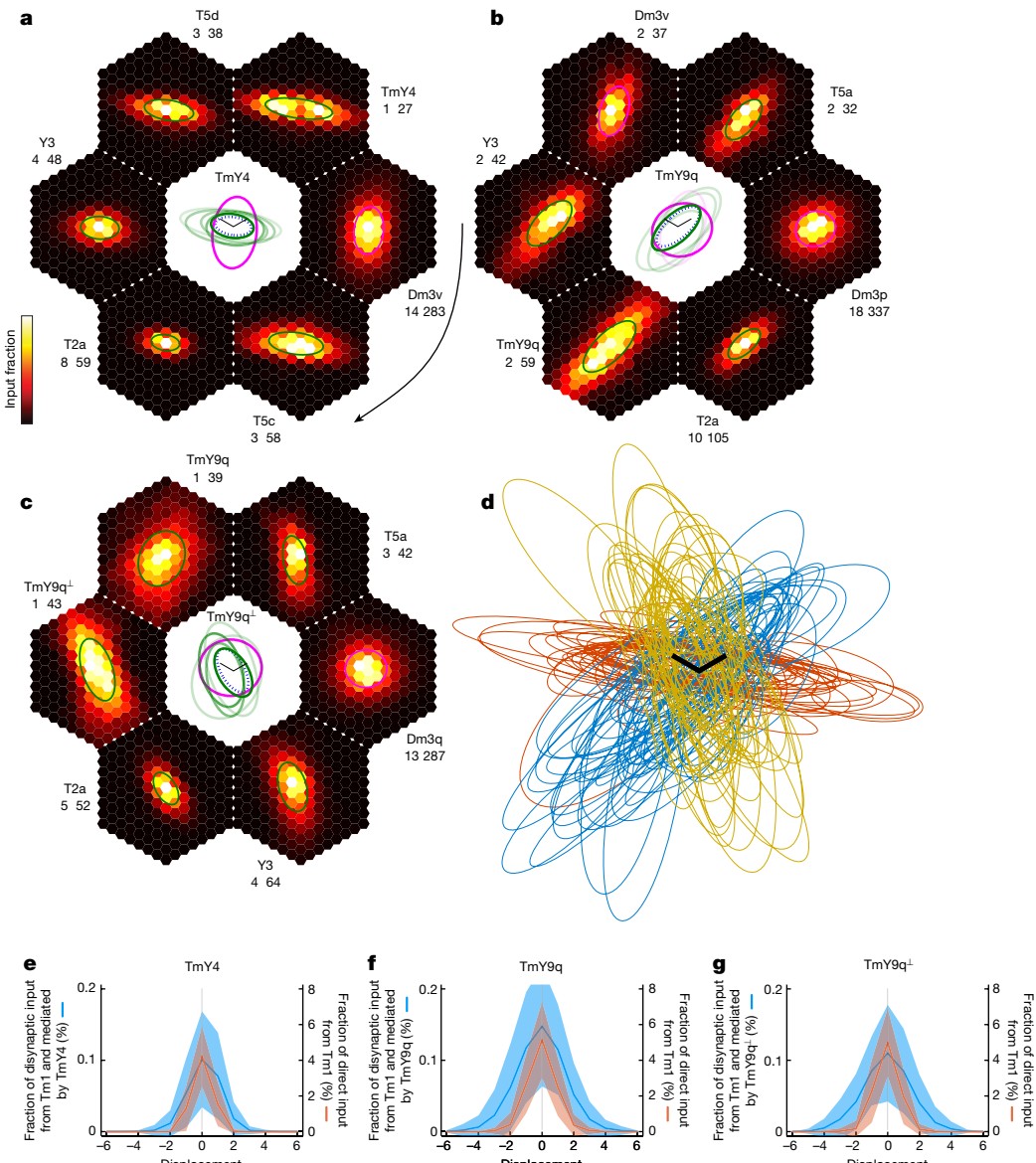

**Fig. 5 | TmY ERFs predicted by mapping disynaptic pathways. a–c,** Heat maps of disynaptic pathways from Tm1 to TmY4, TmY9q and TmY9q$^\perp$, aligned and averaged over target cells. The portion of a disynaptic map outside the pCRF is a predicted ERF component mediated by the named intermediary type (text next to map). Pathways are ranked clockwise (curved arrow) by anatomical strength (Extended Data Fig. 7, green lines). The numbers indicate the fraction of disynaptic input in units of 0.01%: heat map maximum (white) and spatial sum over hexels. For each TmY type, the top pathway is mediated by the orthogonal Dm3 type, and is presumed to be inhibitory. The remaining pathways are presumed to be excitatory. Ellipse approximations are superimposed on each average map, with magenta (green) signifying pathways presumed to be inhibitory (excitatory). Centre: ellipses are shown again, anchored on the average pCRF (dotted ellipse) and enlarged for visibility by ×1.5. The ellipse opacity represents the anatomical strength relative to the strongest inhibitory (magenta) or excitatory (green) pathway. **d,** Aligned ellipse approximations to Tm1–TmY9q–TmY9q (blue), Tm1–TmY9q$^\perp$–TmY9q$^\perp$ (gold) and Tm1–TmY4–TmY4 (brown) disynaptic maps; 50 representative cells each. **e–g,** Disynaptic (Tm1–TmY–TmY) maps extend beyond pCRFs, as shown by 1D longitudinal projections of average maps. A pERF component is predicted in both pCRF end zones. The unit of displacement is lattice constant × $\sqrt{3}$/2. Ribbons represent standard deviation across TmY cells. TmY and Tm1 sample sizes are as in Fig. 2. Scale bars (**a–c** centre, **d**), one lattice constant (*p* and *q* axes).

## TmY extraclassical receptive fields

I predict the ERF of TmY cells by considering disynaptic pathways from Tm1 to TmY types. The strongest pathways are mediated by Dm3 (Extended Data Fig. 7), and the resulting pERF covers mainly the pCRF side zones and is expected to be suppressive (Fig. 5a–c). Such cross-orientation suppression could have the function of sharpening orientation tuning[7].

Of the weaker pERFs that are facilitating (Fig. 5a–c and Extended Data Fig. 7), the TmY–TmY pERF extends the farthest into the pCRF end zones (Fig. 5a–c). Ellipse approximations show the degree of variability across cells (Fig. 5d). 1D projections show that the pERF extends into both end zones (Fig. 5e–g). This pathway could lead to iso-orientation facilitation in the end zones, as the TmY–TmY connections are between the same TmY type.

The extension of the facilitating TmY–TmY pERF into the pCRF end zones is consistent with the idea that TmY–TmY connectivity aids in the completion of noisy or illusory contours. A similar 'collinear facilitation' in visual cortex has been proposed to enable completion of noisy or illusory contours, because an orientation detector receiving weak or ambiguous input from the image can be driven over threshold by excitation from neighbouring orientation detectors[10,11,36].

The TmY–TmY pERFs also extend slightly into the pCRF side zones. This is consistent with the idea that TmY–TmY connectivity can give rise to positional invariance while preserving orientation selectivity[8,9].

pERFs and pCRFs for individual TmY cells are provided in Supplementary Data 4. The geometric relationships evident in the average pERFs do hold for many individual cells.

## Prediction of spatial normalization

The TmY pERF was predicted above by disynaptic pathways terminating at TmY, but trisynaptic or longer pathways could also contribute. Of particular interest is Tm1–TmY–Dm3–TmY, which involves recurrent inhibition of TmY by Dm3. This pathway is expected to contribute a suppressive pERF located in the far side zones of the pCRF and tuned to the preferred orientation (Supplementary Data 4). In the near side zones, this pathway could be obscured by a facilitating pERF due to Tm1–TmY–TmY (Fig. 5a–c and Supplementary Data 4).

A similar excitatory–inhibitory motif is found for many types in the Dm interneuron family and can be viewed as implementing spatial normalization, a common computation in image processing[37]. The inhibitory neuron pools activity of excitatory neurons over some spatial neighbourhood, and sends recurrent inhibition back to the same excitatory neurons. An unusual aspect here is that recurrent inhibition in the Tm1–TmY–Dm3–TmY pathway is opposed by recurrent excitation in the Tm1–TmY–TmY pathway, in the case of side-by-side stimuli at the preferred orientation.

## Pathways leaving the optic lobe

I have predicted that the three Dm3 and three TmY types respond selectively to the local orientation of a visual stimulus. This is conceptualized as a decomposition of the image into small oriented elements such as edges or bars, similar to early visual processing in computer science[38] as well as models of visual cortex[39]. In such models, the computation of local orientation is a general starting point for recognizing all visual forms by subsequent processing steps.

Similarly, information about local stimulus orientation might be used by the fly brain for form vision. TmY cells synapse onto a number of visual projection neurons (VPNs) that project out of the optic lobe into the central brain. I will focus on the most numerous of these VPN types, LC15 and LC10e, which number about 50 cells each (Fig. 6 and Supplementary Data 5). These are of particular interest, because they retain more information about location and are therefore potentially more useful for recognizing complex visual forms. The other VPNs, by contrast, discard more information about location by pooling widely over space, and will be studied elsewhere.

## LC15 invariance to orientation

LC15 neurons project from the lobula to the posterior ventrolateral protocerebrum (PVLP), a neuropil in the central brain[13] (Fig. 6a). LC15 receives input from all three TmY types (Fig. 6a and Extended Data Figs. 4, 5 and 8).

Direct connections from hexel types to LC15 are weak or non-existent, so here the top disynaptic pathways are mapped to predict feature selectivity (Extended Data Fig. 7). Aligning and averaging the maps yields Fig. 6b. Maps for individual cells are in Supplementary Data 5, and are summarized by ellipses in Fig. 6c.

Given that LC15 input from TmY cells of all three orientations is indiscriminate, it is natural to reason that LC15 responds indiscriminately to any orientation, detecting a disjunction (logical OR) of activity in the three channels. If TmY orientation tuning is sufficiently broad, it follows that LC15 should be activated by a stimulus of any orientation. Indeed, recordings of LC15 visual responses show that LC15 is activated by bars of any orientation[40,41].

Other strong intermediaries of disynaptic pathways include T3, Tm25 and Tm21 (Fig. 6b and Extended Data Fig. 7). T3 is known to be a small-object detector[42], and Tm25 and Tm21 may be object detectors on the basis of their connectivity[4]. Indeed, recordings of LC15 responses show that LC15 is also activated by small objects, although not as strongly as by bars[40,41].

## LC10e as possible junction detector

LC10 neurons project from the lobula to the anterior optic tubercle of the central brain[12] (Fig. 6d). LC10 has been divided into four types (LC10a to LC10d) with distinct stratification profiles in the lobula[13]. I identified a new LC10e type, and further subdivided it into dorsal and ventral variants (LC10ev and LC10ed) on the basis of their distinct connectivity patterns (Extended Data Fig. 8). Both variants receive inputs from TmY9 types, but these inputs are stronger for LC10ev (Fig. 6d and Extended Data Fig. 8), so I will focus on it.

Disynaptic pathways starting from hexel types were mapped, and the top intermediaries turned out to be TmY9q and TmY9q$^\perp$ (Fig. 6e and Extended Data Figs. 4d, 5d and 7). The maps were aligned and averaged (Fig. 6e). The average Tm1–TmY9q–LC10ev map is to the left of the average Tm1–TmY9q$^\perp$–LC10ev map (compare green and blue ellipses in Fig. 6e centre). The displacement vectors between ellipse centres (Fig. 6f, inset) are within 90° of the $q$ direction for almost all LC10ev cells. This suggests a systematic spatial relationship between the receptive field components mediated by TmY9q and TmY9q$^\perp$. Noisiness in this relationship is evident in the maps for individual cells (Fig. 6f and Supplementary Data 5), but many individual cells are consistent with the systematic relationship.

Although LC10ev receives input from both TmY9q and TmY9q$^\perp$, it is unclear how these two pathways interact. One possibility is that LC10ev, like LC15, detects a disjunction of activity in two pathways, and responds to a stimulus at either the $q$ or $q^\perp$ orientation. Another possibility is that LC10ev detects a conjunction (logical AND) of activity in the two pathways. In the latter case, LC10ev could be a detector of corners or T-junctions, owing to the systematic spatial relationship evident in Fig. 6e,f. This is just a speculation, because there are additional strong inputs to LC10ev other than TmY9 (Fig. 6e and Supplementary Data 4), and it is unclear how they shape the feature selectivity of LC10ev.

## Motion, object and colour vision

TmY pERFs mediated by T4 and T5 cells (Fig. 5a–c, Extended Data Figs. 4 and 5 and Supplementary Data 3) cover the pCRF end zones, and are facilitating. T4 and T5 neurons are activated by motion, and are also known to prefer oriented stimuli[43]. Therefore, input from the motion system may enhance the orientation selectivity of TmY cells, and also cause TmY responses to be stronger for moving stimuli than for flashed stimuli. Dm3 inputs and outputs do not contain cell types known to encode motion direction (Extended Data Fig. 1).

TmY4 receives inhibition from several lobula plate interneurons, including the full-field cells LPi14 and LPi15 (Extended Data Figs. 4a and 5a), which are presumed to be inhibitory (Methods), and receive strong input from T4a and T5a, and from T4b and T5b, respectively. This suggests that TmY4 might be suppressed by horizontal background motion. TmY9q$^\perp$ (and to a lesser extent TmY9q) receives input from LPi07 (Extended Data Figs. 4c and 5c), suggesting that TmY9q$^\perp$ (and TmY9q) might be suppressed by vertical background motion if LPi07 is inhibitory, because LPi07 receives input from T4c and T5c, and from T4d and T5d.

TmY outputs include motion-related types. TmY4 synapses onto LPi07 cells (Extended Data Figs. 4 and 5), which synapse onto T4c and T5c, and from T4d and T5d. TmY9q and TmY9q$^\perp$ synapse onto a number of types (Y1, Y11, Y12 and so on) that are reciprocally coupled to T4 and T5 cells[4,44].

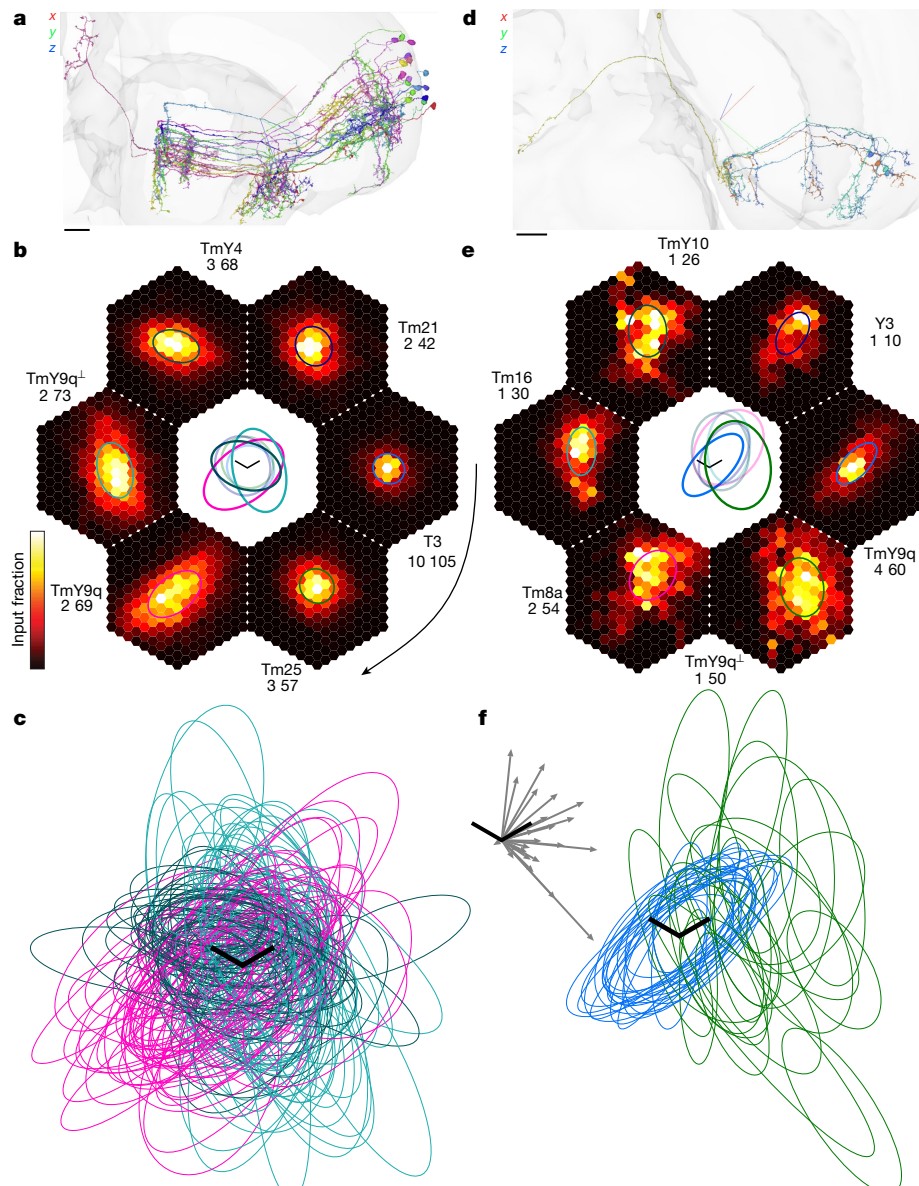

**Fig. 6 | LC15 and LC10e feature selectivity predicted by mapping disynaptic pathways. a**, An LC15 neuron (pink) projecting from the lobula (middle) to the PVLP (upper left), and strong (>4 synapses) presynaptic TmY4, TmY9q and TmY9q$^{\perp}$ partners, with dendrites in the distal medulla (right). **b**, Heat maps of disynaptic pathways from hexel types to LC15, averaged after aligning on Mi1–T3–LC15 maps. Intermediary types (text next to maps) are ranked clockwise (curved arrow) by anatomical strength (Extended Data Fig. 7, blue lines). Maps are shown for the strongest source hexel type: Mi1 for the top two and Tm1 for the rest. The numbers indicate the fraction of disynaptic input in units of 0.01%: heat map maximum (white) and spatial sum over hexels. Ellipse approximations are superimposed on each average map, and shown again in the centre, enlarged ×1.5 for visibility. Ellipses with greater opacity indicate the intermediaries TmY4, TmY9q and TmY9q$^{\perp}$. **c**, Ellipse approximations to Tm1–TmY–LC15 maps for all 54 individual LC15 cells, mediated by TmY4 (dark green), TmY9q (magenta),

and TmY9q$^{\perp}$ (cyan). **d**, An LC10ev neuron (green) projecting from the ventral lobula (middle) to the anterior optic tubercle (far left), and strong (>3 synapses) presynaptic TmY9q and TmY9q$^{\perp}$ partners. **e**, The same as **b**, but for LC10ev, inhibitory intermediaries excluded, averaged after aligning on Tm1–TmY9q–LC10ev maps. Maps are shown for the strongest source hexel type: Tm1, Tm1, Mi9, Mi4, Mi4 and Mi1. Central ellipses with greater opacity indicate the intermediaries TmY9q and TmY9q$^{\perp}$. **f**, Ellipse approximations to Tm1–TmY–LC10ev maps for all 23 individual LC10ev cells, mediated by TmY9q (blue) and TmY9q$^{\perp}$ (green). Inset: displacement vectors from Tm1–TmY9q–LC10ev to Tm1–TmY9q$^{\perp}$–LC10ev map centres are almost all within 90° of the *q* direction. Crosshairs in **a**,**d** are lateral (red), ventral (green) and posterior (blue). Scale bars, 15 μm (**a**), 20 μm (**d**) or one lattice constant along *p* and *q* axes (**b**,**c**,**e**,**f**). For interactive visualizations of the neurons in **a**,**d**, see Supplementary Data 1.

The Dm3–TmY circuit may also be modulated by object-detecting circuitry, given the Dm3 pERFs mediated by T2a (Fig. 4a–c), and the TmY pERFs mediated by T2a and Y3 (Fig. 5a–c). Recordings of T2a and Y3 visual responses have not been reported, but their connectivity (Extended Data Fig. 9) can be used to speculate about function. T2a is similar in connectivity to T3 (ref. 4), which is activated by small objects[42], so T2a may also be a small-object detector[4]. T3 is known to be ON–OFF[42]. T2a and Y3 are likewise predicted to be ON–OFF, because

they receive input from the ON cell Mi1 as well as from Tm1 (Extended Data Fig. 9a,c).

In addition to the predicted ERFs in Figs. 4 and 5, there are other strong pathways to Dm3 and TmY types starting from hexel types other than Tm1 (Extended Data Fig. 7, red and blue lines) and passing through intermediary types such as TmY10 and TmY11 (Extended Data Fig. 10 and Supplementary Data 3 and 4); and disynaptic pathways to LC10e are mediated by Tm8a, TmY10 and Tm16, as well

as TmY9 (Fig. 6e). Recordings of these additional intermediary types have not been reported, but a companion paper uses connectivity to speculate that they are involved in colour vision[4].

## Predictions and their limitations

By interpreting a neuronal wiring diagram, I have predicted visual responses of *Drosophila* neurons that have not previously been recorded by neurophysiologists. Testing these predictions seems bound to be informative, whether or not they turn out to be correct.

CRFs of Dm3 and TmY cells were predicted by mapping monosynaptic pathways from Tm1 cells, and found to be oriented (Figs. 1b–d and 2b–d). On this basis, Dm3 and TmY cells were predicted to be orientation selective.

pERFs were predicted by mapping disynaptic pathways from Tm1 cells (Figs. 4 and 5), which involve connections in the Dm3–TmY circuit (Fig. 3 and Extended Data Fig. 6). A Dm3 pERF mediated by other Dm3 types was predicted to be suppressive in the pCRF side zones, and tuned to non-preferred orientations. A Dm3 pERF mediated by T2a was predicted to be facilitating in the pCRF end zones, and not tuned to the orientation or contrast of the modulating stimulus (Fig. 4a–c).

A TmY pERF mediated by TmY was predicted to be facilitating in the pCRF end zones and near side zones, and tuned to modulating stimuli at the pCRF orientation (Fig. 5a–c). A TmY pERF mediated by Dm3 was predicted to be suppressive in the pCRF side zones and tuned to modulating stimuli orthogonal to the pCRF orientation (Fig. 5a–c). A TmY pERF due to a trisynaptic pathway mediated by both the same TmY type and the orthogonal Dm3 type was predicted to be suppressive in the pCRF side zones for stimuli at the pCRF orientation (Supplementary Data 4).

Although my predictions are powerfully concrete and specific, they also have limitations. First, I relied on the fundamental assumption that the CRF arises from monosynaptic connectivity, and that the ERF arises from polysynaptic pathways that are not paralleled by a monosynaptic connection. One can be confident about the assumption for an inhibitory pathway such as Tm1–Dm3v–TmY4, which by itself should be incapable of activating the TmY4 cell. However, stimulating an excitatory pathway such as Tm1–TmY4–TmY4 could conceivably activate the final TmY4 cell, even if the direct Tm1–TmY4 pathway is not stimulated. In general, my fundamental assumption is more certain for inhibitory and less certain for excitatory polysynaptic pathways.

If experiments show that the size and shape of CRFs match the predictions of Figs. 1 and 2, then the fundamental assumption will be upheld, and further experiments can proceed to look for the predicted facilitating and suppressive ERF components.

On the other hand, if the CRF turns out to be longer than predicted, that would mean excitatory polysynaptic pathways, here predicted to contribute to the ERF, instead turn out to lengthen the CRF. The prediction of a facilitating ERF in the CRF end zone would become invalid, because the end zone would be swallowed up by the lengthened CRF. The ERF would then be determined solely by inhibitory polysynaptic pathways and be only suppressive.

The above uncertainty is related to some ambiguity inherent in the ERF definition[5]. It is supposed to be impossible to evoke a neural response through stimulation of the ERF alone (without CRF stimulation). If the definition is revised to substitute difficult for impossible, the distinction between ERF and CRF is no longer black and white.

In the second limitation, predictions can be challenging owing to the existence of many polysynaptic pathways that can contribute to the ERF (Figs. 4 and 5). When relevant pathways are both excitatory and inhibitory, it can be difficult to predict the overall sign (facilitating versus suppressing) of modulation by a stimulus outside the CRF. I have accordingly focused my ERF predictions on locations where either excitation or inhibition is likely to dominate.

However, there are additional ERF pathways that were not considered above. For example, a companion paper[37] predicts numerous normalization mechanisms that are presynaptic to the Dm3–TmY circuit: Tm1 responses are spatially normalized by five Pm interneuron types that pool over multiple length scales (Extended Data Fig. 2), L2 (the dominant input to Tm1) is normalized by three Dm interneuron types, and the L2–Tm1 connection is normalized by two Dm interneuron types. All of these mechanisms are expected to contribute suppressive ERF components that are not tuned to stimulus orientation, and could obscure the predicted facilitating ERF components.

A third limitation concerns variation across cells of the same type, which is summarized by the ellipses in Figs. 1h, 2j, 4d, 5d, 6c,f and detailed in Supplementary Data 3–5. A systematic kind of variation is the distortion of geometric relationships near the borders of the eye. There is also apparently random variation even in the interior of the eye, far from the borders. pERFs tend to look more variable (Figs. 4d and 5d and Supplementary Data 3 and 4) than pCRFs (Figs. 1 and 2), which may be due in part to randomness in the coverage of the visual field by non-hexel types.

Finally, my conjecture about LC10e assumes a high threshold for activation, which suggests that LC10e should detect a conjunction of TmY9q and TmY9q$^\perp$ activation, triggered by two oriented stimuli arranged in a corner or T-junction (Fig. 6e,f and Supplementary Data 4). Alternatively, LC10e might detect a disjunction of its inputs, if it has a low threshold for activation. LC15 is indiscriminately connected with all three TmY types, and its observed invariance to stimulus orientation[40,41] suggests that LC15 has a low threshold for activation. An additional subtlety, not further discussed here, is that threshold actually varies dynamically owing to the presence of inhibitory inputs.

In principle, some of the above limitations could be overcome by building network models of neural activity. That might not improve the predictive power at this time, however, because so many model parameters are unknown at present. Synapse counts are only crude estimates of physiological connection strength[14], and unitary conductance can differ markedly between inhibitory and excitatory synapses[45]. Other uncertainties, such as whether inhibition should be modelled as subtractive or divisive[46], go beyond parameters. Therefore, it is crucial to carry out experiments that reduce model uncertainty. If visual physiologists measure orientation and location tuning curves, that would constrain the strength of cross-orientation inhibition relative to direct Tm1 excitation, as well as the relative strengths of ERF pathways. Biophysical experiments would clarify the extent to which voltage varies throughout a neuronal arbour[47–49], which is important because pCRF and pERF pathways may terminate in synapses at different locations on the arbour.

## Discussion

Convolutional nets are a popular approach to form vision in artificial intelligence. Optic lobe cell types can be interpreted as feature maps in a convolutional net[50], and average Tm1–Dm3 (Fig. 1b–d) and Tm1–TmY (Fig. 2d–f) connectivity maps can be interpreted as convolution kernels. LC10e and LC15 combine inputs from TmY cells, and may be regarded as complex feature detectors built by combining inputs from simpler feature detectors. This accords with the original hierarchical conception of convolutional nets[39], which was inspired by hypothetical wiring diagrams for primary visual cortex (V1) in mammals[31].

The wiring of the Dm3–TmY circuit resembles connectivity motifs originally proposed for V1 (refs. 8,34,35), and some visual responses predicted here for fly neurons have precedents in V1. In particular, a 'collinear facilitation' is predicted for the ERF of TmY cells, meaning that a stimulus of the preferred orientation in the end zone can facilitate the response to a stimulus of the preferred orientation in the CRF. Collinear facilitation was previously reported for V1 neurons[10,11,36],

but most reports emphasize suppression[6]. The conflicting reports might be reconciled if ERFs are generated by diverse parallel polysynaptic pathways[6]. Different visual stimuli might emphasize either excitatory or inhibitory pathways, leading to different experimental results.

The ERFs of TmY cells are similarly predicted to be shaped by many overlapping excitatory and inhibitory pathways (Figs. 4 and 5 and Supplementary Data 4), but the complexity is constrained by the connectome. We can reason about the cell types and connections (as in this work), observe and manipulate them with the aid of genetic tools, and use them to build neural network models. This new opportunity has the potential to overcome the limitations of traditional methods used to disentangle the effects of overlapping ERF mechanisms in the cortex, such as manipulating stimulus contrast, orientation and timing[6,51].

'Line amacrine' cells probably homologous to Dm3 have been described in several dipteran species[1]. If Dm3 and TmY homologues turn out to exist in honeybees, it will be worth comparing with studies of honeybee behaviour, with the caveat that visual acuity is higher and inter-ommatidial spacing is lower in honeybees than in *Drosophila*[52].

Behavioural experiments were interpreted to infer the existence of "at least three orientation-sensitive channels" in the bee visual system[53,54]. Orientation-selective neurons were later predicted to span a maximum of three ommatidia[55]. This roughly matches the predicted CRFs of Dm3 and TmY cells. It is worth noting that Dm3 receptive fields would not previously have been guessed to be so short, because Dm3 dendrites are quite long (Fig. 1b–d). The predicted CRFs require true connectivity, not just morphology.

The collinear facilitation predicted for TmY cells could have consequences for insect visual behaviours. Bees were once claimed to perceive illusory contours[56,57]. The experiments were later declared to be irreproducible[58,59], although this belated renunciation seems to have escaped notice[60]. The topic seems worth reopening, if neurophysiology experiments can detect the collinear facilitation predicted here. The design of illusory contour stimuli seems more likely to be successful if guided by neurophysiology as well as behaviour.

Behavioural experiments with honeybees were used to argue that the orientation of a visual stimulus is computed independently from its direction of motion[53]. Orientation selectivity was subsequently discovered in direction-selective T4 and T5 cells of *Drosophila*[43]. The present work predicts an orientation-selective Dm3–TmY circuit that is indeed distinct from the motion circuit, although there are interactions between them as described above.

The Dm3–TmY circuit immediately suggests many questions about neural development. Unknowns include the identities of the molecules that guide Dm3 dendrites to grow in three directions, and how the Dm3 dendrites 'decide' how far to grow, even after turning sharply at the border of the medulla (Fig. 1). The molecules that establish connectivity preferences of Dm3 and TmY cells (Fig. 3a) are of obvious interest. There should also be molecules that establish preferences for synapse formation at particular dendritic locations, which are important for the spatial organization of connectivity (Extended Data Fig. 6) and remain to be analysed in future work.

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

## Methods

### Reconstructed wiring diagram of a female adult fly brain

This work is based on v783 of the proofread reconstruction of an adult female *Drosophila* brain[15–17]. All cells and connections are drawn from the right optic lobe. Cell-type annotations of neurons intrinsic to the optic lobe[4], and 'boundary' neurons that straddle the optic lobe and central brain[17] are provided in companion papers.

As detailed elsewhere[16,61], neurons were reconstructed by human proofreading of a 3D electron micrograph that was automatically segmented using convolutional nets. Synapses were automatically detected and assigned to partners using convolutional nets[62]. Reconstruction accuracy is state-of-the-art, judging from comparisons[4,16] with neuronal wiring diagrams in the fly central brain[63] and optic lobe[64] previously reconstructed by other methods. I have not tried to disentangle biological and technical sources of cell-to-cell variability, although this could be attempted using statistical models[17,64].

It is common to threshold the wiring diagram by retaining only connections with at least some threshold number of synapses[16,17,63]. This thresholding is carried out to reduce false-positive connections, and is important in the central brain, where most cell types consist of just a single neuron and its mirrored counterpart in the other hemisphere. In the optic lobe, most types contain many cells, so a large sample can be used to decide whether cell types are genuinely connected. Hence, the present study does not threshold connections.

Annotations of Dm3, TmY4 and TmY9 cell types were released with a companion paper[4] but were mentioned only in passing. The present work describes connectomic properties of these cell types in detail for the first time.

### Dm3 types

Line amacrine cells were described in Strausfeld's Golgi studies of *Calliphora* and *Eristalis*[1], as well as *Musca*[65]. Strausfeld also mentioned unpublished observations of line amacrine cells in *Locusta* and *Apis*[1]. Line amacrine cells were named Dm3 in a Golgi study of *Drosophila*[2].

Light microscopy with multicolour stochastic labelling[3] went beyond Golgi studies by splitting Dm3 into two types with dendrite at orthogonal orientations. Dm3p and Dm3q were then shown to have transcriptomes that differ before adulthood (P50 or earlier)[18]. (Ref. 18 used the alternative names Dm3a and Dm3b.) Immunostaining showed that Dm3q expresses Bifid, whereas Dm3p does not. Ref. 18 also analysed a reconstruction of seven medulla columns[64], with the results showing that Dm3p and Dm3q prefer to synapse onto each other, foreshadowing the present work, and speculatively placed Dm3 cells in the motion pathway.

It is unclear why this approach did not find the third Dm3v type, which is obvious in Fig. 1d. Presumably it is because only Dm3p and Dm3q were labelled in the GAL4 line used by the study. There was no way to see Dm3v because it was invisible. Subsequent transcriptomic studies[18,19] did not uncover the third Dm3 type. Neither did the seven-column medulla electron microscopy dataset[18,64] uncover it, presumably because this volume is too small to contain more than fragments of Dm3 cells. However, the three Dm3 types (Fig. 1b–d) are unmistakably obvious in our complete and unbiased sample of optic lobe neurons. They can be distinguished by the directions of their neurites (Fig. 1b–d), or by their patterns of connectivity (Fig. 3a and Extended Data Fig. 1).

In connectivity patterns, Dm3p and Dm3q are more similar to each other than to Dm3v, and TmY9q and TmY9q$^\perp$ are more similar to each other than to TmY4[4]. Whether similarity of connectivity corresponds with transcriptomic similarity remains to be seen.

### TmY types

TmY4 and TmY9 were previously described[2]. The two TmY9 types can be distinguished by the tangential directions of their neurites (Fig. 2b,c), or by their connectivity (Fig. 3a, and Extended Data Figs. 4 and 5). Their stratification profiles are slightly different (Fig. 1a). TmY9q$^\perp$ stratifies in layers 1 and 2 of the lobula plate, whereas TmY9q stratifies only in layer 1. TmY9q$^\perp$ is more often bistratified in layers 5 and 6 of the lobula, whereas TmY9q is more often monostratified (Figs. 1a and 2b,c).

### LC10e and LC15

LC10 cells project from the lobula to the anterior optic tubercle[12], and have been linked with visually guided courtship behaviours[66]. Four LC10 types were previously identified using GAL4 transgenic lines, on the basis of their stratification in the lobula[13]. Using the connectomic approach described in a companion paper[4], I identified a fifth type (LC10e), which stratifies in layer 6 of the lobula. LC10e was further subdivided into two groups on the basis of connectivity. The two groups cover the dorsal and ventral medulla, respectively.

My conjecture that LC10e detects a corner or T-junction is specific to the ventral variant, which receives strong input from TmY9q and TmY9q$^\perp$. The ventral visual field is expected to be more important for form vision, assuming that the fly is above the landmarks or objects to be seen.

### Neurotransmitter and receptor identity

FlyWire provides predictions of neurotransmitter identity that are based on the electron micrographs[67]. Dm3 is predicted to be glutamatergic, whereas TmY4 and TmY9 are predicted to be cholinergic. The same inferences can be drawn by examining expression of neurotransmitter synthesis and transport genes[19,68].

Whether a neurotransmitter has an excitatory or inhibitory effect on the postsynaptic neuron depends on the identity of the postsynaptic receptor. Acetylcholine is excitatory when the postsynaptic receptor is nicotinic, which is generally the case in the fly brain[69]. Glutamate is inhibitory in *Drosophila* when the postsynaptic receptor is GluClα[70].

According to transcriptomic data[19,68], Dm3 expresses GluClα. Unpublished data indicate that TmY4 and TmY9 also express GluClα (Y. Kurmangaliyev, personal communication). It should be noted that transcriptomic information so far exists for Dm3p and Dm3q, but not Dm3v.

On the basis of the above evidence, Dm3 is presumed to be inhibitory whereas TmY4 and TmY9 are presumed to be excitatory in the present work.

Tm1, Tm2, Tm9 and L3 are predicted to be cholinergic, and Mi4 is predicted to be GABAergic by FlyWire[67] and transcriptomics[68].

T2a, Y3, TmY10, TmY11, Tm7, Tm8a, Tm16, Tm20, Tm25 and Tm27 are predicted to be cholinergic on the basis of electron micrographs[16,67], and presumed to be excitatory.

LPi14 and LPi15 are predicted to be GABAergic on the basis of electron micrographs[16,67], and presumed to be inhibitory. LPi07 cells are predicted to be GABAergic, glutamatergic or uncertain on the basis of electron micrographs, and are presumed to be inhibitory.

### Hexel cell types

The ommatidia of an insect compound eye are typically organized into a hexagonal lattice, so the term hexel will refer to an element of the image captured by the compound eye. This is to distinguish the geometry from an image defined on a square lattice of pixels, as customary in computer vision.

The *Drosophila* compound eye obeys the principle of neural superposition, in which each hexel is sampled by six photoreceptors that are located in neighbouring ommatidia but point at the same optical axis[2,71]. These six photoreceptors converge onto a single lamina cartridge, which projects to a single medulla column.

Cell types that occur once per cartridge or column are said to be modular[64], and are in one-to-one correspondence with hexels. I define hexel cell types as those that are modular, and also have receptive fields that have been observed by neurophysiologists to be approximately one ommatidium wide. Tm1, Tm2, Tm9, Mi1, Mi4 and Mi9 are

included because the full-width at half-maximum of their receptive fields ranges from 6° to 8°, roughly equivalent to the angular spacing between ommatidia[30]. L1 to L5 are also included, on the basis of observed receptive fields[33]. (L3 turns out to be the main contributor to the disynaptic pathways studied.) This list of hexel types is provisional because receptive fields of modular types have not yet been quantified exhaustively.

Connectivity maps included 745 Tm1, 746 Tm2, 716 Tm9, 796 Mi1, 749 Mi4, 730 Mi9, 785 L1, 763 L2, 709 L3, 671 L4 and 774 L5 cells in the v783 reconstruction that were successfully assigned to points in the hexagonal lattice through the procedures explained below. These numbers are smaller than the total number of cells proofread in v783 (ref. 4), but the deficit is generally less than 10%.

Tm3 and Tm4 were excluded from the list of hexel types because the full-width at half-maximum of their receptive fields is 12° (ref. 30). Including Tm3 and Tm4 would change the list of strong disynaptic pathways. Tm4–Li02–LC10ev, for example, would enter the list for LC10ev. T4 and T5 were also not considered hexel types because their receptive fields are too large.

Tm1 is OFF transient[30,32,72], Tm2 is OFF transient[30,32,72,73], Mi4 is ON sustained[30,74], Tm9 is OFF sustained[30,32,72], and L3 is OFF sustained[33,75]. These are all Dm3 inputs (Extended Data Figs. 1 and 3), consistent with the prediction that Dm3 cells have OFF receptive fields. Mi1 is ON transient[30,72,76]. Most of these physiological studies are based on calcium imaging. Electrophysiology[72] and voltage imaging[48] are also possible.

## Hexagonal lattice coordinates

Rules of connectivity in the optic lobe were simplified in a companion paper[4] to depend on only cell type, and neglected spatial locations. For a refined characterization of optic lobe connectivity that includes spatial dependences, the present work assigned hexel cell types to a hexagonal lattice.

All Mi1 cells were semi-automatically assigned to hexagonal lattice points. Locations of L cells were assigned by placing them in one-to-one correspondence with Mi1 cells using the Hungarian algorithm applied to the connectivity matrix. The locations of other hexel types were assigned by placing them in one-to-one correspondence with L cells, again with the Hungarian algorithm.

The resulting locations of hexel types in $(p, q)$ coordinates are provided in Supplementary Data 2. Following the convention defined in ref. 29, all three cardinal axes of the hexagonal lattice point upwards (Fig. 1f). The vertical axis is directed dorsally. The $p$ and $q$ axes are directed in the anterodorsal and posterodorsal directions in the medulla, respectively. Note that for *Drosophila* the hexagons of the lattice are oriented with flat sides at the top and bottom, and pointy tips at the left and right. The relation of $p$–$q$ axes to dorsoventral and anteroposterior axes is more complex than indicated in Fig. 1f, because the medulla is curved rather than flat.

The figures portray the lattice of medulla columns. A similar lattice can be constructed for ommatidia, and this lattice is left–right inverted relative to the lattice of medulla columns owing to the optic chiasm. Therefore, back-to-front motion on the retina is front-to-back motion on the medulla lattice. In other words, the $p$ and $q$ axes are swapped in the eye relative to the medulla. Another difference between the eye and the medulla is that the $p$ and $q$ axes are close to orthogonal in the medulla, which is squashed along the anterior–posterior direction. The $p$ and $q$ axes are closer to 120° apart in the eye, where the ommatidia more closely approximate a regular hexagonal lattice.

Hexagonal lattices are drawn in the figures as if they were perfectly uniform. The drawings are intended to portray only the nearest-neighbour relations of cells and columns, and do not accurately represent distances. More geometrically accurate representations of the lattices were constructed in ref. 29, which quantitatively characterized how lattice properties vary in space for the left optic lobe of the same electron microscopy dataset used in this study, and for many

*Drosophila* eyes[29]. Visual acuity also varies across the retina in flies and other insects[52,77].

## Centres of 'receptive fields'

Locations for Dm3 and TmY cells were computed from maps of monosynaptic connectivity from Tm1. Locations for LC15 and LC10ev cells were computed from their strongest disynaptic pathways, Mi1–T3–LC15 and Tm1–TmY9q–LC10ev. In all cases, the map was convolved with a linear filter that was 1.1 in the central column and 1 in its six neighbouring columns. The maximum of the result was taken as the location of the cell. This centre is often close to the ellipse centre (the centroid of the map), but they are not necessarily the same.

## Ellipse approximation to 'receptive fields'

Suppose that an image has hexel values $h_i$ at Cartesian coordinates $(x_i, y_i)$, where $i$ runs from 1 to $N$ points of a hexagonal lattice. Normalizing the image yields a probability distribution $p_i = h_i/(\sum_{j=1}^{N} h_j)$. Then compute the coordinates of the image centroid by

$$\bar{x} = \sum_{i=1}^{N} p_i x_i \quad \bar{y} = \sum_{i=1}^{N} p_i y_i$$

and the covariance matrix by

$$C = \sum_{i=1}^{N} p_i \begin{bmatrix} (x_i - \bar{x})^2 & (x_i - \bar{x})(y_i - \bar{y}) \\ (x_i - \bar{x})(y_i - \bar{y}) & (y_i - \bar{y})^2 \end{bmatrix} + \frac{5s^2}{12} I$$

in which $I$ denotes the $2 \times 2$ identity matrix and $s$ denotes the length of a hexagon side. The length and width of the hexel image are defined as $2\sigma_{max}$ and $2\sigma_{min}$, in which $\sigma_{max}^2$ and $\sigma_{min}^2$ are the larger and smaller eigenvalues of the covariance matrix. The approximating ellipse is centred at the image centroid, and oriented along the principal eigenvector of the covariance matrix.

The first term of the covariance matrix $C$ effectively regards the probability distribution as a weighted combination of Dirac delta functions located at the lattice points. The second term is a correction that arises if each delta function is replaced by a uniform distribution over the corresponding hexagon. This replacement makes biological sense because a column receives visual input from a non-zero solid angle. Without the correction, the length and width would vanish if the image consists of a single hexel concentrated at a single delta function. With the correction, the length and width of an image with a single non-zero hexel become $s\sqrt{5/3} = a\sqrt{5}/3$, in which $a = s\sqrt{3}$ is the lattice constant. The correction becomes relatively minor when the length and width of the image are large.

The above has implicitly defined $2\sigma$ as the width of a 1D Gaussian distribution, for which $\sigma^2$ is the variance. This is the full-width at $e^{-1/2} \approx 0.6$ of the maximum. Alternatively, the width could be estimated by the full-width at half-maximum, $\sigma 2\sqrt{2\ln2} \approx 2.4\sigma$. For either estimate, the width is proportional to $\sigma$. I stick with the simpler estimate $2\sigma$, which can be readily scaled by any multiplicative factor of the reader's preference.

## 1D projections of 'receptive fields'

The ellipse approximation gives a parametric estimate of receptive field size. For a non-parametric estimate, I also used 1D projections onto directions defined on the hexagonal lattice. Each hexel was given coordinates $(p, q)$, with the origin placed at the anchor location used for alignment.

The cardinal directions ($v$, $p$ and $q$) point to nearest neighbours, which are one lattice constant away. The projection grouped hexels with equal $p + q$, $2p - q$ or $2q - p$. The projection was smoothed by convolving with [0.5, 1, 0.5] with stride 2, resulting in coordinates measured in units of a single lattice constant.

The orthogonal directions ($h$, $p^\perp$ and $q^\perp$) point to next-nearest neighbours, which are $\sqrt{3}$ lattice constant away. The projections grouped hexels with equal $q-p$, $q$ or $p$. The resulting coordinate was in units of lattice constant $\times\sqrt{3}/2$.

## Input fractions of cell and cell types

Let $W_{ab}$ be the number of synapses from cell $a$ to cell $b$, and let $W_{AB}$ be the number of synapses from cell type $A$ to cell type $B$. Normalize these matrices so that every column sums to 1,

$$P_{ab} = \frac{W_{ab}}{\sum_c W_{cb}} \quad P_{AB} = \frac{W_{AB}}{\sum_C W_{CB}}$$

The matrix $P_{ab}$ is the fraction of input synapses to neuron $b$ that come from neuron $a$. Similarly, the matrix $P_{AB}$ is the fraction of input synapses to cell type $B$ that come from cell type $A$.

The matrix $P_{ab}$ can be interpreted as the Markov chain defined by a 'backwards' random walk on neurons. At each time step, the walker chooses an input synapse of the present neuron uniformly at random, and then crosses the synapse in the retrograde direction to reach the presynaptic node. Then $P_{ab}$ denotes the probability of stepping from neuron $b$ to neuron $a$.

Similarly, the matrix $P_{AB}$ can be interpreted as the Markov chain defined by a 'backwards' random walk on cell types. At each time step, the walker chooses an input synapse of the present cell type uniformly at random, and then crosses the synapse in the retrograde direction to reach the presynaptic cell type. Then $P_{AB}$ denotes the probability of stepping from cell type $B$ to cell type $A$.

Extended Data Fig. 7 quantifies the anatomical strength of a disynaptic pathway by $P_{AB}P_{BC}$, which is the probability of randomly walking the $A$–$B$–$C$ pathway backwards. This score was used to select the top disynaptic pathways leading to Dm3 (Fig. 4), TmY (Fig. 5) and LC (Fig. 6) types.

## Intermediary types

The analysed disynaptic pathways (Figs. 4–6, Extended Data Fig. 7 and Supplementary Data 3–5) start at a hexel type, pass through an intermediary type, and finish at a target type (Dm3, TmY or LC). The intermediary type is constrained to not be a hexel type. For LC10ev (Fig. 6 and Supplementary Data 5), the intermediary type is constrained to be cholinergic (excitatory). For LC15, the constraint was unnecessary as the top intermediaries are all predicted to be cholinergic (excitatory). The analyses do not require that intermediary types be assigned spatial locations. The number of cells for each intermediary type can be found in a companion paper[4].

## Mapping monosynaptic connectivity

Let $\mathbf{r}_a$ denote the location of cell $a$ in the hexagonal lattice, and let $W_{ab}$ denote the number of synapses from cell $a$ to cell $b$. Then the monosynaptic connectivity map from cell type $A$ to cell $b$ is

$$f_{Ab}(\mathbf{r}) = \sum_{a\in A} \delta_{\mathbf{r}\mathbf{r}_a} W_{ab}$$

The average monosynaptic connectivity map from cell type $A$ to cell type $B$ is

$$f_{AB}(\mathbf{r}) = \frac{1}{|B|} \sum_{a\in A}\sum_{b\in B} \delta_{\mathbf{r},\mathbf{r}_a-\mathbf{r}_b} W_{ab}$$

The normalization is with respect to the number of cells $|B|$ of type $B$, because the connectivity map is defined as the average number of synapses received by a $B$ cell from $A$ cells, if the origin of the coordinate system is placed at the $B$ cell.

In Supplementary Data 3–5, these maps are defined using $P_{ab}$ rather than $W_{ab}$, to facilitate comparison with the probability maps for disynaptic pathways defined below.

## Mapping disynaptic pathways

The map of the disynaptic pathway from cell type $A$ to cell type $B$ to cell $c$ is

$$f_{ABc}(\mathbf{r}) = \sum_{a\in A}\sum_{b\in B} \delta_{\mathbf{r}\mathbf{r}_a} P_{ab} P_{bc}$$

and the average disynaptic pathway map from cell type $A$ to cell type $B$ to cell type $C$ is

$$f_{ABC}(\mathbf{r}) = \frac{1}{|C|} \sum_{a\in A}\sum_{b\in B}\sum_{c\in C} \delta_{\mathbf{r},\mathbf{r}_a-\mathbf{r}_c} P_{ab} P_{bc}$$

Both maps are normalized probability distributions, in the sense that summing over all $\mathbf{r}$, $A$ and $B$ yields 1.

Maps for trisynaptic pathways are defined in an analogous fashion.

## Reporting summary

Further information on research design is available in the Nature Portfolio Reporting Summary linked to this article.

## Data availability

Cell-type annotations for the optic lobe were taken from visual_neuron_types.csv.gz (6 May 2024), and those for the central brain were taken from classification.csv.gz (6 March 2024). Neurotransmitter annotations were taken from neurons.csv.gz (27 February 2024). These files are downloadable from the FlyWire Codex (https://codex.flywire.ai). Assignments of hexel types to hexagonal lattice points are provided in Supplementary Data 2.

## Code availability

Figures can be reproduced using code available at https://github.com/hsseung/OpticLobe.jl.

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

**Acknowledgements** I thank K. Kruk for typing virtually all of the cells used in this analysis[4]; E. Kind for sharing his annotations of Mi1 locations in the right optic lobe; R. Willie for illustrating the neuropil layers in Fig. 1a; A. Seung, N. Seung and S. M. Seung for helping to assign Mi1 cells to hexagonal lattice points; Y. Kurmangaliyev and M. Murthy for comments on the manuscript; Y. Yin for corrections to an earlier version; C. Gilbert for advice about collinear facilitation; T. Currier, T. Clandinin and A. Matsliah for conversations; and J. Maitin-Shepard for Google's Neuroglancer. This research was supported by the National Institutes of Health (awards U24NS120053, U19NS132720, RF1MH129268, U24NS126935, RF1MH123400, UM1NS132250 and UM1NS132253).

**Competing interests** The author has a financial interest in Zetta AI.

**Additional information**
**Correspondence and requests for materials** should be addressed to H. Sebastian Seung.

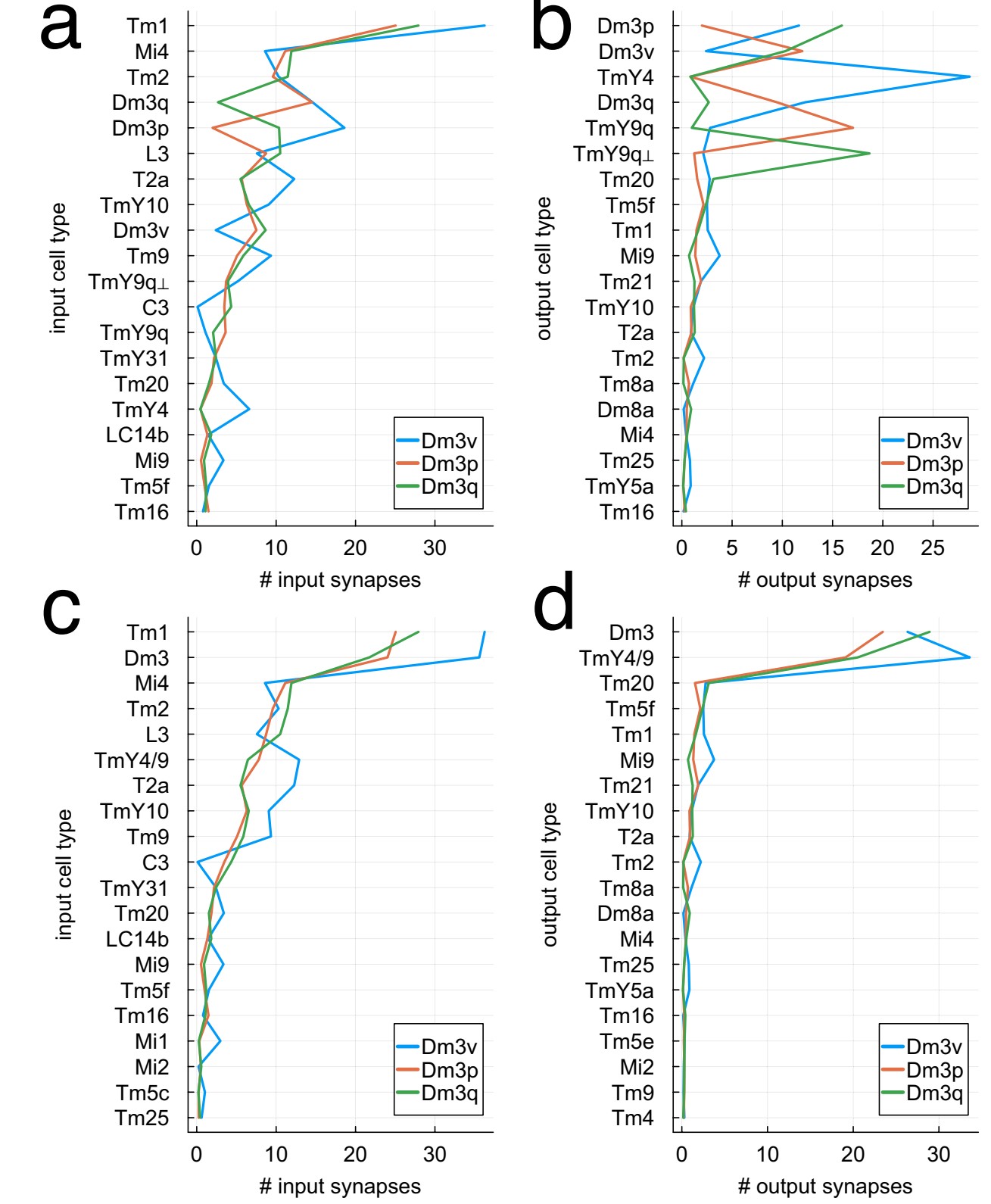

**Extended Data Fig. 1 | Dm3 inputs and outputs. a**, Average number of input synapses received by a Dm3 cell from input cell types, which are ordered by the total number of synapses they make onto all three Dm3 types. **b**, Average number of output synapses sent by a Dm3 cell to output cell types, which are ordered by the total number of synapses they receive from all three Dm3 types. **c**, Same as (**a**), except Dm3v, Dm3p, and Dm3q are grouped into the presynaptic class Dm3, and TmY4, TmY9q, and TmY9q⊥ are grouped into the presynaptic class TmY4/TmY9. **d**, Same as (**b**), except for the groupings Dm3 and TmY4/TmY9.

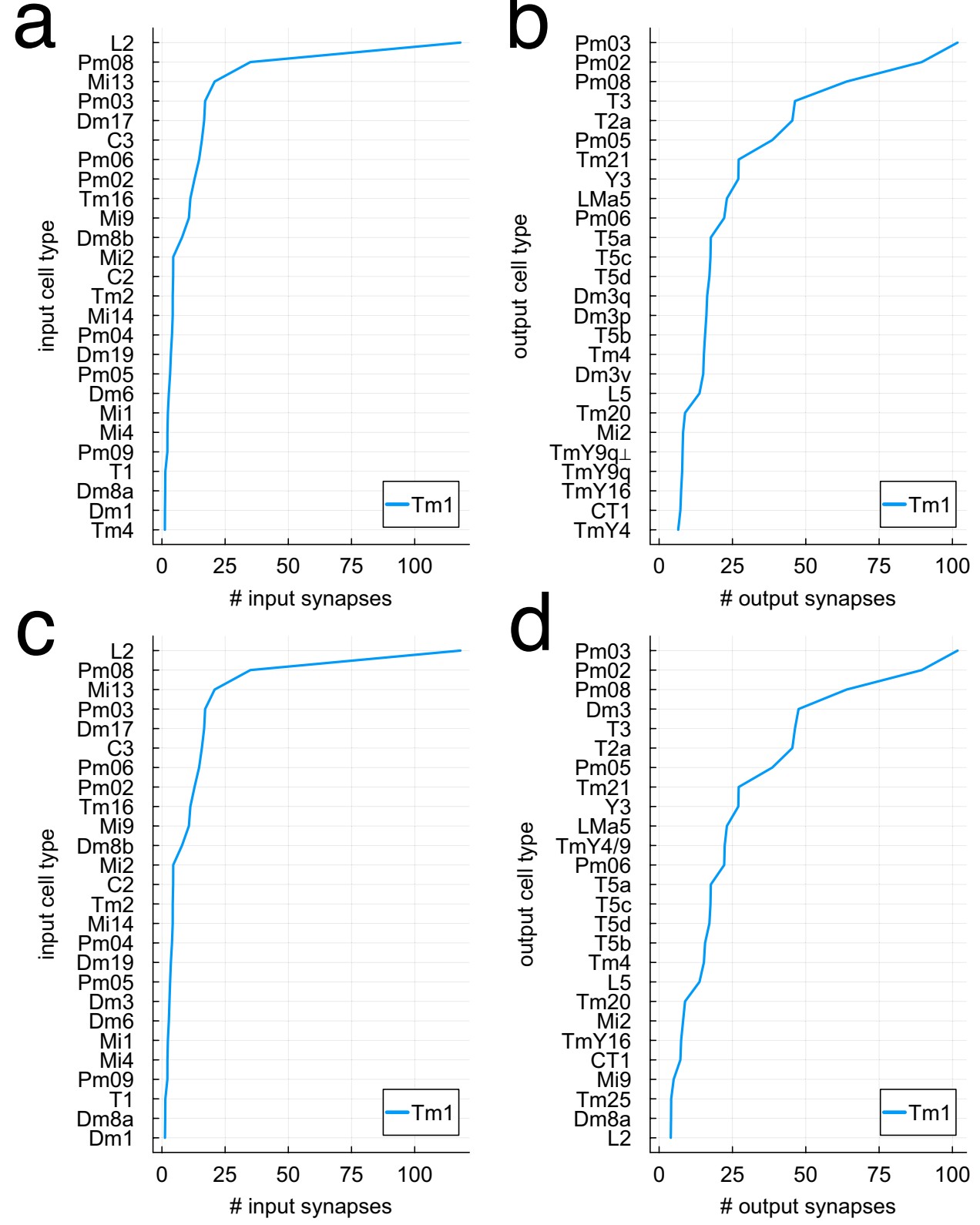

**Extended Data Fig. 2 | Tm1 inputs and outputs. a**, Average number of input synapses received by a Tm1 cell from presynaptic cell types. **b**, Average number of output synapses sent by a Tm1 cell to postsynaptic cell types. **c**, Same as (**a**), except Dm3v, Dm3p, and Dm3q are grouped into the presynaptic class Dm3, and TmY4, TmY9q, and TmY9q⊥ are grouped into the presynaptic class TmY4/ TmY9. **d**, Same as (**b**), except for the groupings Dm3 and TmY4/TmY9.

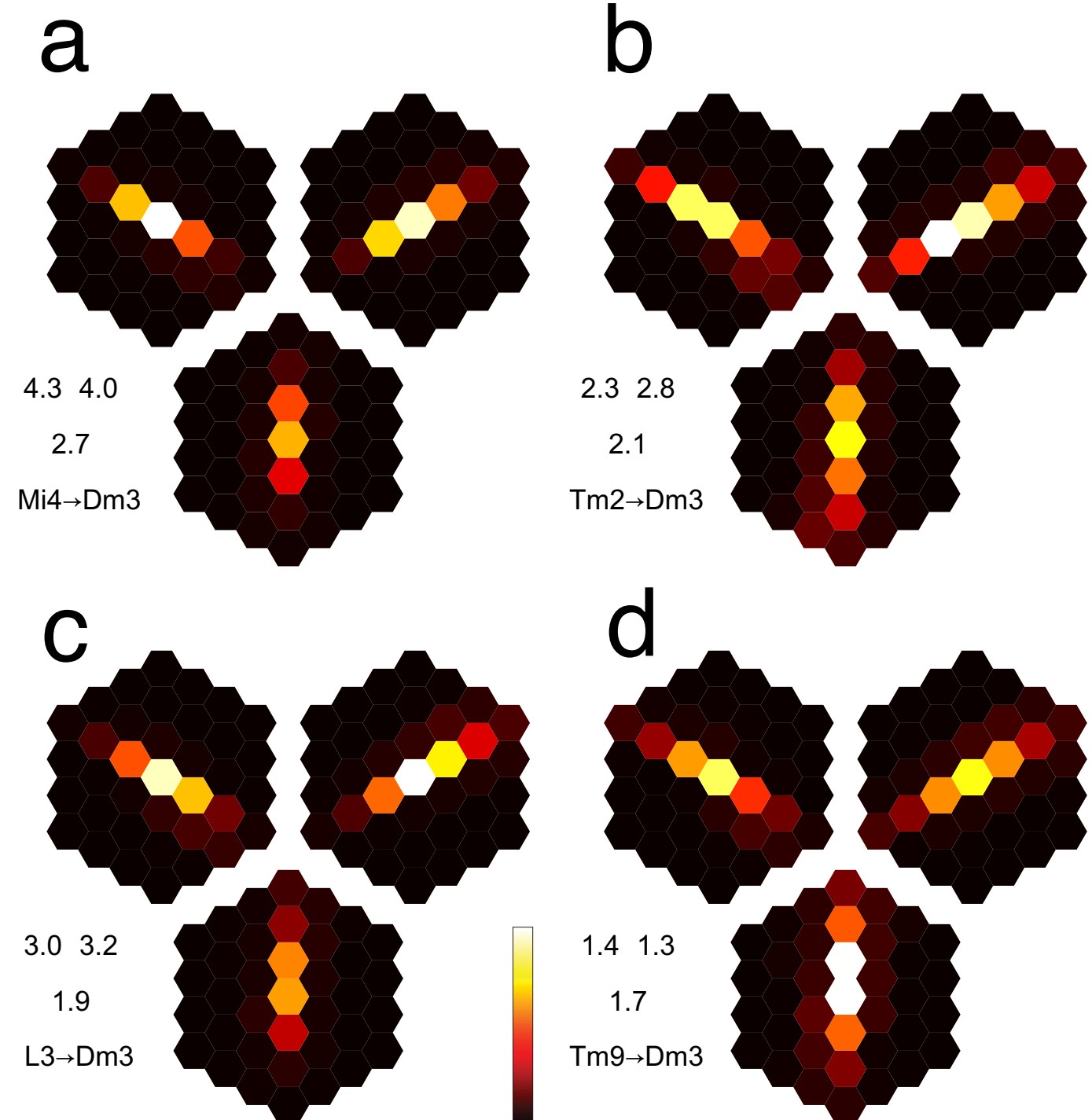

**Extended Data Fig. 3 | Average connectivity maps from hexel types to Dm3.** Connectivity maps of the hexel types most strongly connected (after Tm1) to Dm3, after aligning and averaging. The presynaptic hexel types are Mi4 (**a**), Tm2 (**b**), L3 (**c**), and Tm9 (**d**), and the postsynaptic Dm3 types in each panel are Dm3v (bottom), Dm3p (top left), and Dm3q (top right). Maps are in units of synapses per hexel. For each panel, the maximum values (white) of the color map are indicated by the trio of numbers at left, and the minimum value is zero. The location of each Dm3 cell was computed from the Tm1-Dm3 connectivity maps (Methods), so the average maps expose any systematic spatial offsets relative to the Tm1-Dm3 maps. The L3-Dm3v map, for example, is shifted upward by roughly half a hexel. Dm3 cell numbers are given in Fig. 1.

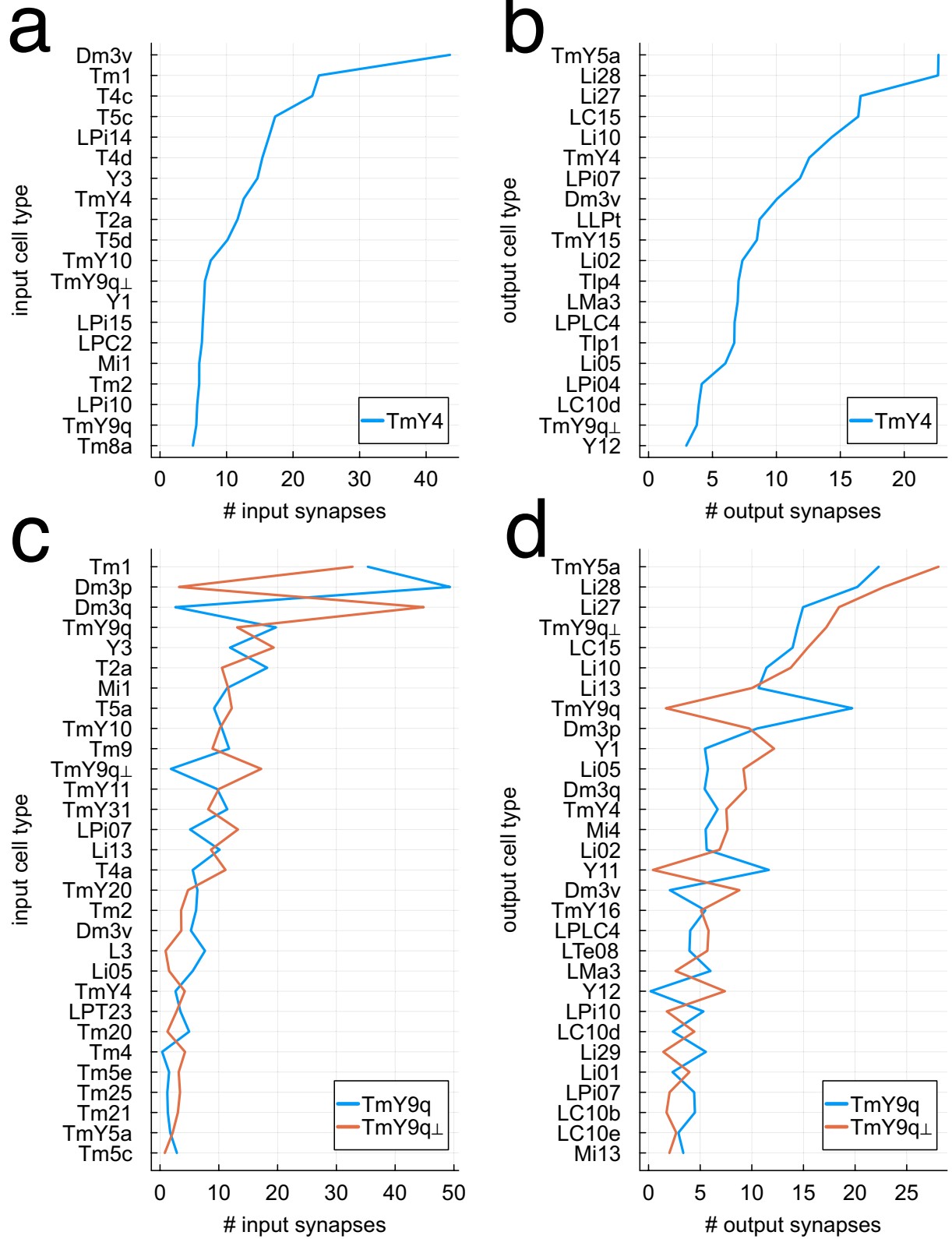

**Extended Data Fig. 4 | TmY4 and TmY9 inputs and outputs. a**, Average number of input synapses received by a TmY4 cell from presynaptic cell types. **b**, Average number of output synapses sent by a TmY4 cell to postsynaptic cell types. **c**, Average number of input synapses received by a TmY9 cell from presynaptic cell types. **d**, Average number of output synapses sent by a TmY9 cell to postsynaptic cell types.

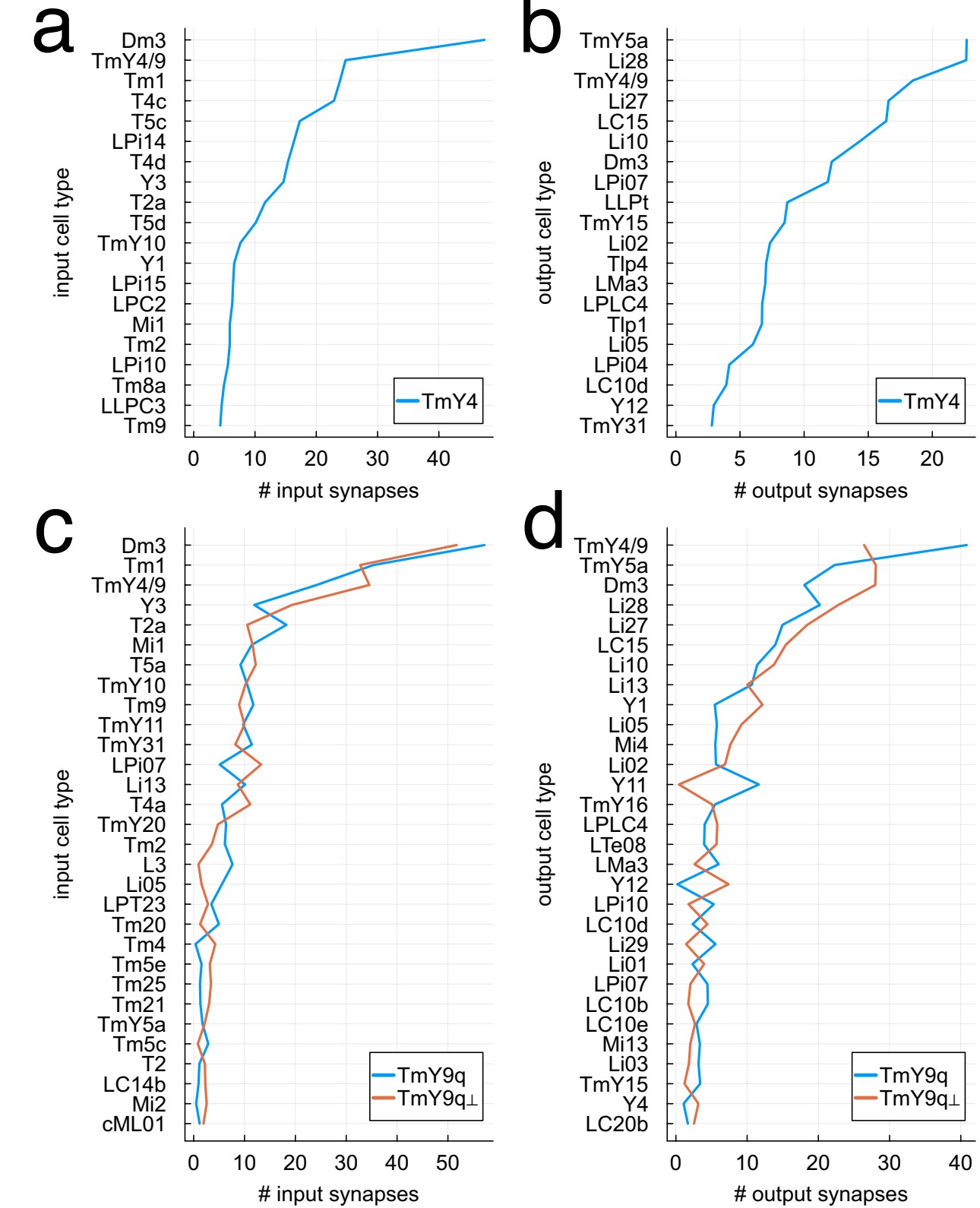

**Extended Data Fig. 5 | TmY4 and TmY9 inputs and outputs after groupings.**
Same as Extended Data Fig. 4, except Dm3v, Dm3p, and Dm3q are grouped into the presynaptic class Dm3, and TmY4, TmY9q, and TmY9q⊥ are grouped into the presynaptic class TmY4/TmY9. **a**, Average number of input synapses received by a TmY4 cell from presynaptic cell types. **b**, Average number of output synapses sent by a TmY4 cell to postsynaptic cell types. **c**, Average number of input synapses received by a TmY9 cell from presynaptic cell types. **d**, Average number of output synapses sent by a TmY9 cell to postsynaptic cell types.

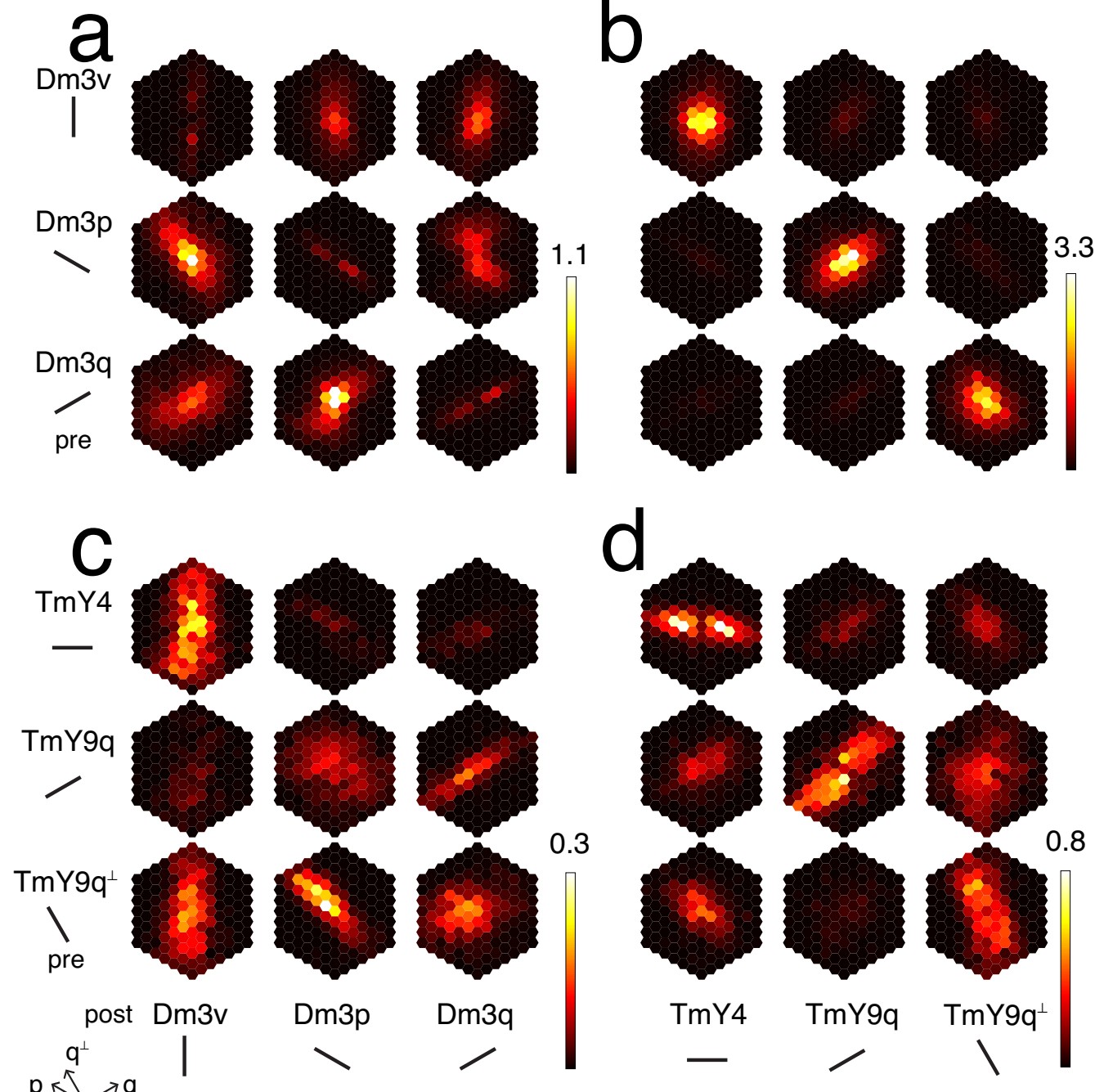

**Extended Data Fig. 6 | Spatial organization of Dm3 and TmY lateral connections. a**, Average Dm3–Dm3 connectivity maps. Each hexel represents the average number of synapses from a presynaptic cell at that location, assuming the postsynaptic cell is located at the central hexel. The location of each cell is computed from its Tm1 inputs (Methods). **b**, Average Dm3–TmY connectivity maps. **c**, Average TmY–Dm3 connectivity maps. **d**, Average TmY–TmY connectivity maps.

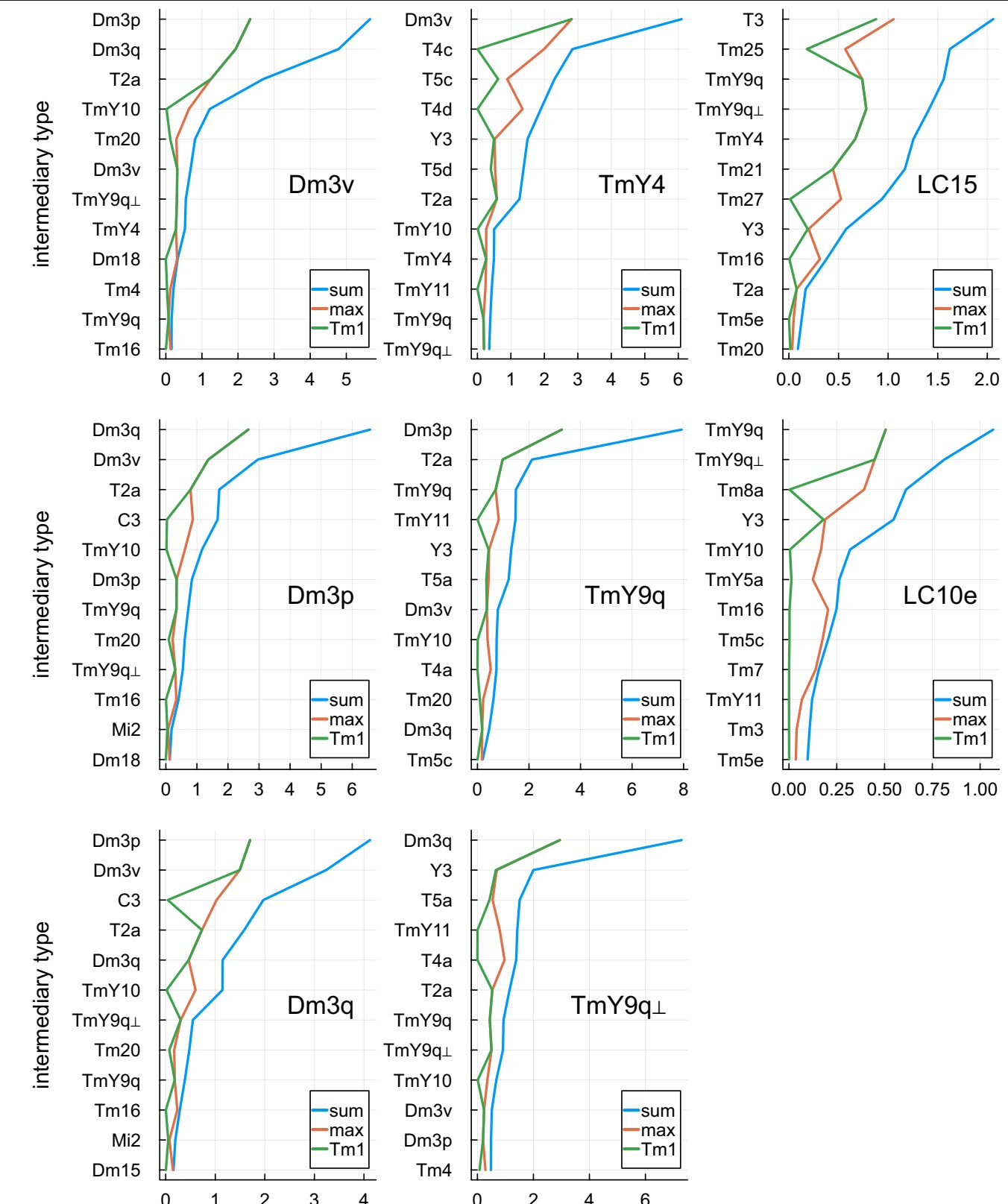

**Extended Data Fig. 7 | Strength estimates for disynaptic pathways.**
Strengths of disynaptic pathways from hexel type to intermediary type to Dm3, TmY, and LC target types (Methods). Intermediary types are constrained to not be hexel types. For each target, intermediary types are ranked by summing strengths over pathways that share the same intermediary type and target type but start from different hexel types (blue). Strengths are also plotted for the largest term in the sum (red), and for the pathway from Tm1 to intermediary type to target type (green). Units are percent.

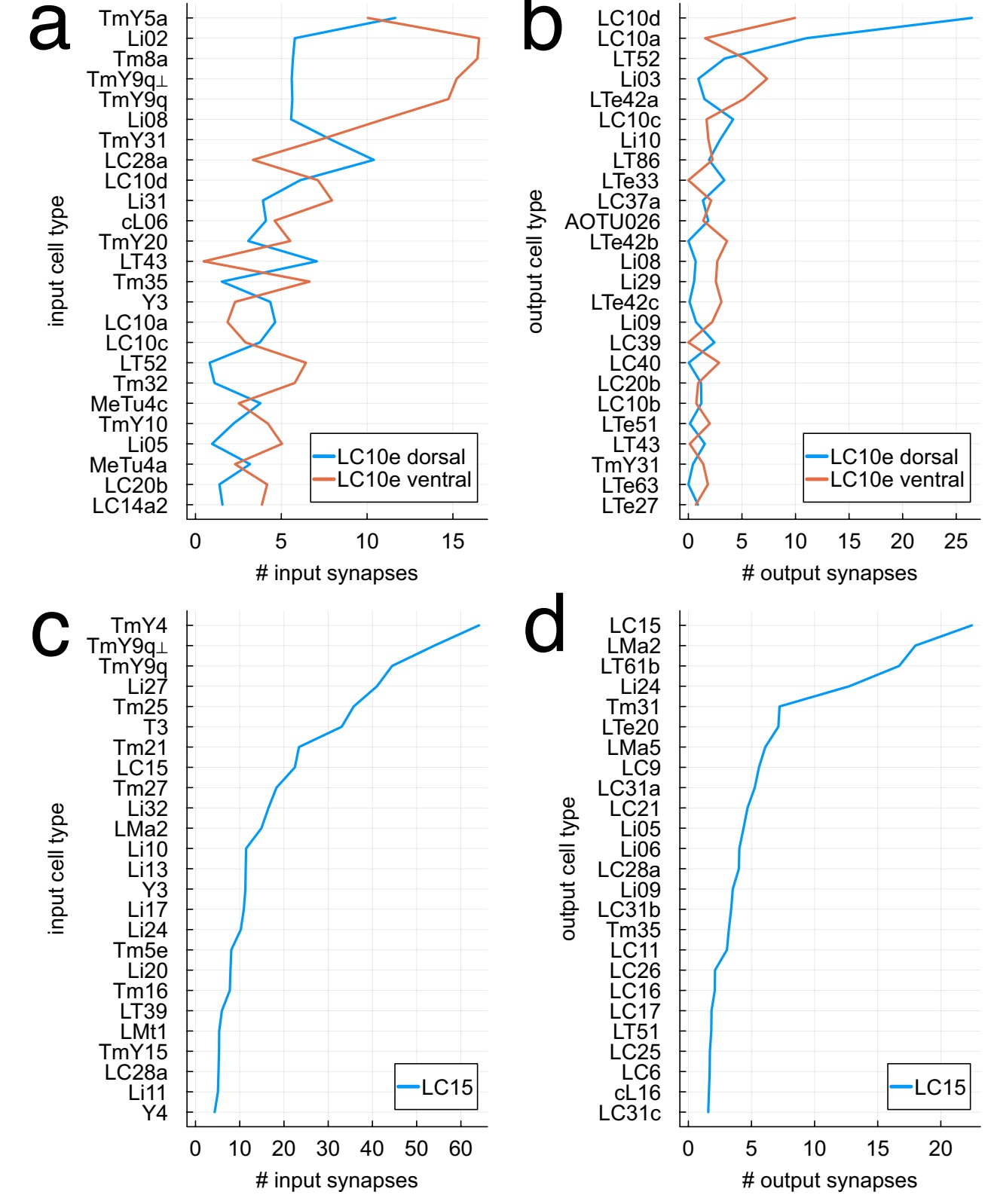

**Extended Data Fig. 8 | LC10e and LC15 inputs and outputs.** Partner cells are restricted to optic lobe neurons (intrinsic or boundary types), and do not include central brain neurons. **a**, Average number of input synapses received by an LC10e cell from presynaptic cell types. **b**, Average number of output synapses sent by an LC10e cell to postsynaptic cell types. **c**, Average number of input synapses received by an LC15 cell from presynaptic cell types. **d**, Average number of output synapses sent by a LC15 cell to postsynaptic cell types.

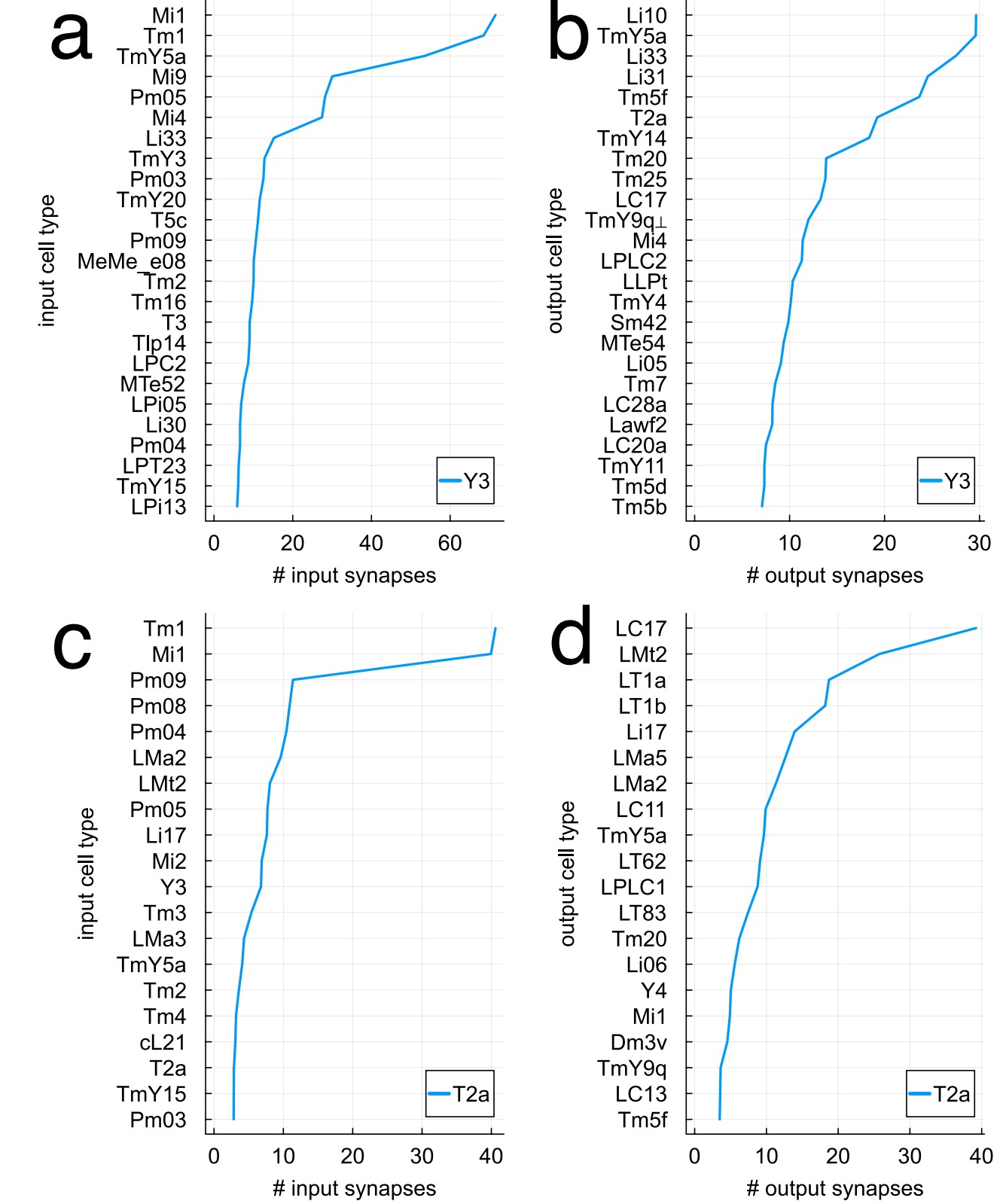

**Extended Data Fig. 9 | Y3 and T2a inputs and outputs. a**, Average number of input synapses received by a Y3 cell from presynaptic cell types. **b**, Average number of output synapses sent by a Y3 cell to postsynaptic cell types.

**c**, Average number of input synapses received by a T2a cell from presynaptic cell types. **d**, Average number of output synapses sent by a T2a cell to postsynaptic cell types.

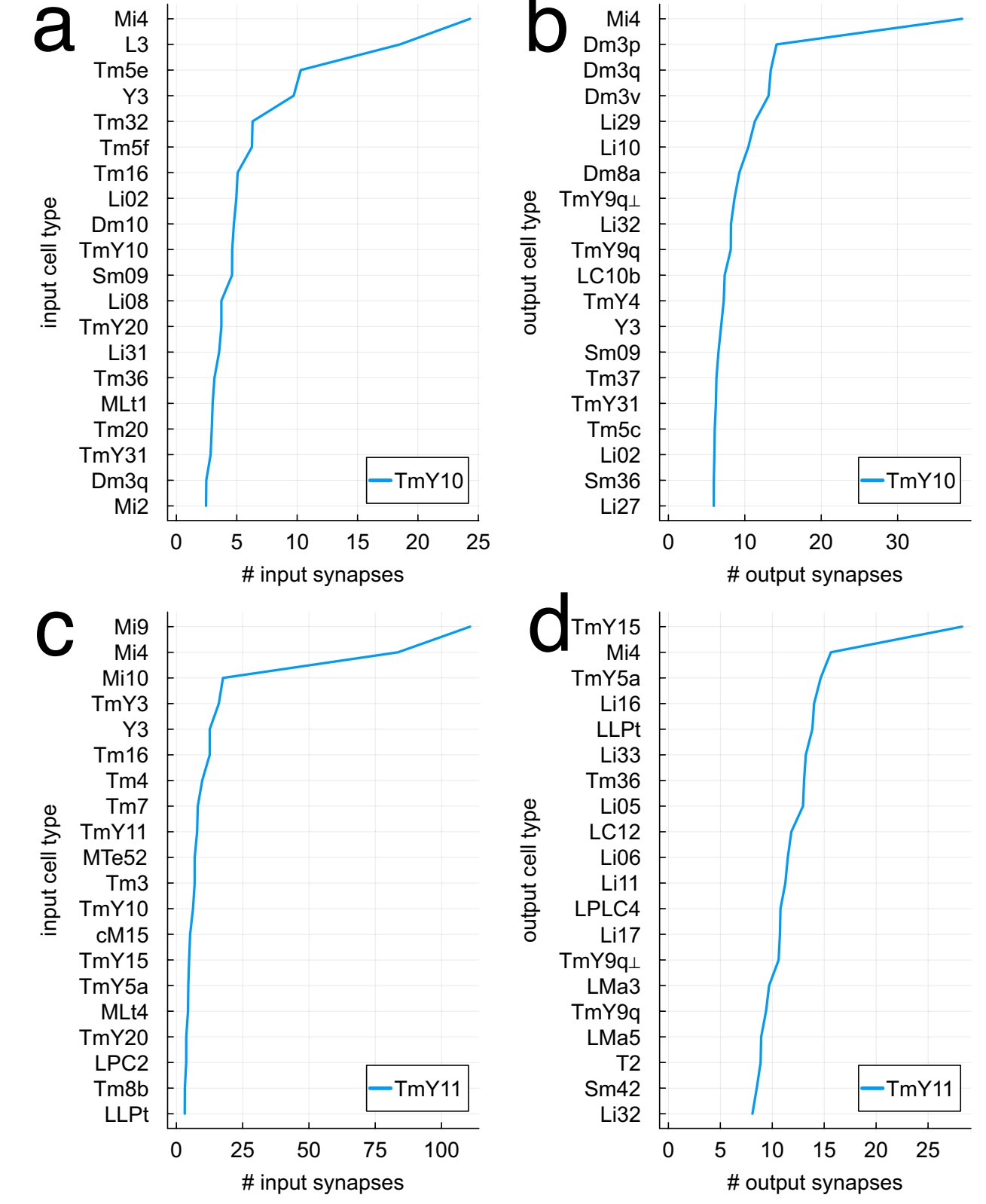

**Extended Data Fig. 10 | TmY10 and TmY11 inputs and outputs. a**, Average number of input synapses received by a TmY10 cell from presynaptic cell types. **b**, Average number of output synapses sent by a TmY10 cell to postsynaptic cell types. **c**, Average number of input synapses received by a TmY11 cell from presynaptic cell types. **d**, Average number of output synapses sent by a TmY11 cell to postsynaptic cell types.

# Reporting Summary

## Statistics

For all statistical analyses, confirm that the following items are present in the figure legend, table legend, main text, or Methods section.

| n/a | Confirmed | |
|---|---|---|
| ☐ | ☒ | The exact sample size (*n*) for each experimental group/condition, given as a discrete number and unit of measurement |
| ☒ | ☐ | A statement on whether measurements were taken from distinct samples or whether the same sample was measured repeatedly |
| ☒ | ☐ | The statistical test(s) used AND whether they are one- or two-sided<br>*Only common tests should be described solely by name; describe more complex techniques in the Methods section.* |
| ☒ | ☐ | A description of all covariates tested |
| ☒ | ☐ | A description of any assumptions or corrections, such as tests of normality and adjustment for multiple comparisons |
| ☐ | ☒ | A full description of the statistical parameters including central tendency (e.g. means) or other basic estimates (e.g. regression coefficient) AND variation (e.g. standard deviation) or associated estimates of uncertainty (e.g. confidence intervals) |
| ☒ | ☐ | For null hypothesis testing, the test statistic (e.g. *F*, *t*, *r*) with confidence intervals, effect sizes, degrees of freedom and *P* value noted<br>*Give P values as exact values whenever suitable.* |
| ☒ | ☐ | For Bayesian analysis, information on the choice of priors and Markov chain Monte Carlo settings |
| ☒ | ☐ | For hierarchical and complex designs, identification of the appropriate level for tests and full reporting of outcomes |
| ☒ | ☐ | Estimates of effect sizes (e.g. Cohen's *d*, Pearson's *r*), indicating how they were calculated |

*Our web collection on statistics for biologists contains articles on many of the points above.*

## Software and code

Policy information about availability of computer code

| Data collection | *Provide a description of all commercial, open source and custom code used to collect the data in this study, specifying the version used OR state that no software was used.* |
|---|---|
| Data analysis | Code for reproducing figures will be deposited in a community repository on or before publication. |

For manuscripts utilizing custom algorithms or software that are central to the research but not yet described in published literature, software must be made available to editors and reviewers. We strongly encourage code deposition in a community repository (e.g. GitHub). See the Nature Portfolio guidelines for submitting code & software for further information.

## Data

Policy information about availability of data

All manuscripts must include a data availability statement. This statement should provide the following information, where applicable:
- Accession codes, unique identifiers, or web links for publicly available datasets
- A description of any restrictions on data availability
- For clinical datasets or third party data, please ensure that the statement adheres to our policy

Primary data is available at the FlyWire Codex (codex.flywire.ai). Derived data has been deposited at https://zenodo.org/records/11388746

## Research involving human participants, their data, or biological material

Policy information about studies with [human participants or human data](). See also policy information about [sex, gender (identity/presentation), and sexual orientation]() and [race, ethnicity and racism]().

| | |
|---|---|
| Reporting on sex and gender | *Use the terms sex (biological attribute) and gender (shaped by social and cultural circumstances) carefully in order to avoid confusing both terms. Indicate if findings apply to only one sex or gender; describe whether sex and gender were considered in study design; whether sex and/or gender was determined based on self-reporting or assigned and methods used.*<br>*Provide in the source data disaggregated sex and gender data, where this information has been collected, and if consent has been obtained for sharing of individual-level data; provide overall numbers in this Reporting Summary. Please state if this information has not been collected.*<br>*Report sex- and gender-based analyses where performed, justify reasons for lack of sex- and gender-based analysis.* |
| Reporting on race, ethnicity, or other socially relevant groupings | *Please specify the socially constructed or socially relevant categorization variable(s) used in your manuscript and explain why they were used. Please note that such variables should not be used as proxies for other socially constructed/relevant variables (for example, race or ethnicity should not be used as a proxy for socioeconomic status).*<br>*Provide clear definitions of the relevant terms used, how they were provided (by the participants/respondents, the researchers, or third parties), and the method(s) used to classify people into the different categories (e.g. self-report, census or administrative data, social media data, etc.)*<br>*Please provide details about how you controlled for confounding variables in your analyses.* |
| Population characteristics | *Describe the covariate-relevant population characteristics of the human research participants (e.g. age, genotypic information, past and current diagnosis and treatment categories). If you filled out the behavioural & social sciences study design questions and have nothing to add here, write "See above."* |
| Recruitment | *Describe how participants were recruited. Outline any potential self-selection bias or other biases that may be present and how these are likely to impact results.* |
| Ethics oversight | *Identify the organization(s) that approved the study protocol.* |

Note that full information on the approval of the study protocol must also be provided in the manuscript.

# Field-specific reporting

Please select the one below that is the best fit for your research. If you are not sure, read the appropriate sections before making your selection.

☒ Life sciences  ☐ Behavioural & social sciences  ☐ Ecological, evolutionary & environmental sciences

For a reference copy of the document with all sections, see [nature.com/documents/nr-reporting-summary-flat.pdf](nature.com/documents/nr-reporting-summary-flat.pdf)

# Life sciences study design

All studies must disclose on these points even when the disclosure is negative.

| | |
|---|---|
| Sample size | Statistics are descriptive rather than hypothesis-testing. |
| Data exclusions | No data was excluded. |
| Replication | One of the basic findings, the identification of a third Dm3 type and the splitting of TmY9 into two types, was originally done in the right optic lobe, and has now been replicated in the left optic lobe. We expect further replication as other fly connectomes become available. |
| Randomization | Statistics are descriptive rather than hypothesis-testing. |
| Blinding | Statistics are descriptive rather than hypothesis-testing. |

# Reporting for specific materials, systems and methods

We require information from authors about some types of materials, experimental systems and methods used in many studies. Here, indicate whether each material, system or method listed is relevant to your study. If you are not sure if a list item applies to your research, read the appropriate section before selecting a response.

## Materials & experimental systems

| n/a | Involved in the study |
|-----|----------------------|
| ☐ ☐ | Antibodies |
| ☐ ☐ | Eukaryotic cell lines |
| ☐ ☐ | Palaeontology and archaeology |
| ☐ ☐ | Animals and other organisms |
| ☐ ☐ | Clinical data |
| ☐ ☐ | Dual use research of concern |
| ☐ ☐ | Plants |

## Methods

| n/a | Involved in the study |
|-----|----------------------|
| ☐ ☐ | ChIP-seq |
| ☐ ☐ | Flow cytometry |
| ☐ ☐ | MRI-based neuroimaging |

## Antibodies

| | |
|---|---|
| Antibodies used | *Describe all antibodies used in the study; as applicable, provide supplier name, catalog number, clone name, and lot number.* |
| Validation | *Describe the validation of each primary antibody for the species and application, noting any validation statements on the manufacturer's website, relevant citations, antibody profiles in online databases, or data provided in the manuscript.* |

## Eukaryotic cell lines

Policy information about cell lines and Sex and Gender in Research

| | |
|---|---|
| Cell line source(s) | *State the source of each cell line used and the sex of all primary cell lines and cells derived from human participants or vertebrate models.* |
| Authentication | *Describe the authentication procedures for each cell line used OR declare that none of the cell lines used were authenticated.* |
| Mycoplasma contamination | *Confirm that all cell lines tested negative for mycoplasma contamination OR describe the results of the testing for mycoplasma contamination OR declare that the cell lines were not tested for mycoplasma contamination.* |
| Commonly misidentified lines (See ICLAC register) | *Name any commonly misidentified cell lines used in the study and provide a rationale for their use.* |

## Palaeontology and Archaeology

| | |
|---|---|
| Specimen provenance | *Provide provenance information for specimens and describe permits that were obtained for the work (including the name of the issuing authority, the date of issue, and any identifying information). Permits should encompass collection and, where applicable, export.* |
| Specimen deposition | *Indicate where the specimens have been deposited to permit free access by other researchers.* |
| Dating methods | *If new dates are provided, describe how they were obtained (e.g. collection, storage, sample pretreatment and measurement), where they were obtained (i.e. lab name), the calibration program and the protocol for quality assurance OR state that no new dates are provided.* |

☐ Tick this box to confirm that the raw and calibrated dates are available in the paper or in Supplementary Information.

| | |
|---|---|
| Ethics oversight | *Identify the organization(s) that approved or provided guidance on the study protocol, OR state that no ethical approval or guidance was required and explain why not.* |

Note that full information on the approval of the study protocol must also be provided in the manuscript.

## Animals and other research organisms

Policy information about studies involving animals; ARRIVE guidelines recommended for reporting animal research, and Sex and Gender in Research

| | |
|---|---|
| Laboratory animals | *For laboratory animals, report species, strain and age OR state that the study did not involve laboratory animals.* |
| Wild animals | *Provide details on animals observed in or captured in the field; report species and age where possible. Describe how animals were caught and transported and what happened to captive animals after the study (if killed, explain why and describe method; if released, say where and when) OR state that the study did not involve wild animals.* |
| Reporting on sex | *Indicate if findings apply to only one sex; describe whether sex was considered in study design, methods used for assigning sex. Provide data disaggregated for sex where this information has been collected in the source data as appropriate; provide overall* |

| | |
|---|---|
| | *numbers in this Reporting Summary. Please state if this information has not been collected. Report sex-based analyses where performed, justify reasons for lack of sex-based analysis.* |
| Field-collected samples | *For laboratory work with field-collected samples, describe all relevant parameters such as housing, maintenance, temperature, photoperiod and end-of-experiment protocol OR state that the study did not involve samples collected from the field.* |
| Ethics oversight | *Identify the organization(s) that approved or provided guidance on the study protocol, OR state that no ethical approval or guidance was required and explain why not.* |

Note that full information on the approval of the study protocol must also be provided in the manuscript.

# Clinical data

Policy information about clinical studies
All manuscripts should comply with the ICMJE guidelines for publication of clinical research and a completed CONSORT checklist must be included with all submissions.

| | |
|---|---|
| Clinical trial registration | *Provide the trial registration number from ClinicalTrials.gov or an equivalent agency.* |
| Study protocol | *Note where the full trial protocol can be accessed OR if not available, explain why.* |
| Data collection | *Describe the settings and locales of data collection, noting the time periods of recruitment and data collection.* |
| Outcomes | *Describe how you pre-defined primary and secondary outcome measures and how you assessed these measures.* |

# Dual use research of concern

Policy information about dual use research of concern

## Hazards

Could the accidental, deliberate or reckless misuse of agents or technologies generated in the work, or the application of information presented in the manuscript, pose a threat to:

No | Yes

- ☐ ☐ Public health
- ☐ ☐ National security
- ☐ ☐ Crops and/or livestock
- ☐ ☐ Ecosystems
- ☐ ☐ Any other significant area

## Experiments of concern

Does the work involve any of these experiments of concern:

No | Yes

- ☐ ☐ Demonstrate how to render a vaccine ineffective
- ☐ ☐ Confer resistance to therapeutically useful antibiotics or antiviral agents
- ☐ ☐ Enhance the virulence of a pathogen or render a nonpathogen virulent
- ☐ ☐ Increase transmissibility of a pathogen
- ☐ ☐ Alter the host range of a pathogen
- ☐ ☐ Enable evasion of diagnostic/detection modalities
- ☐ ☐ Enable the weaponization of a biological agent or toxin
- ☐ ☐ Any other potentially harmful combination of experiments and agents

# Plants

| | |
|---|---|
| Seed stocks | *Report on the source of all seed stocks or other plant material used. If applicable, state the seed stock centre and catalogue number. If plant specimens were collected from the field, describe the collection location, date and sampling procedures.* |
| Novel plant genotypes | *Describe the methods by which all novel plant genotypes were produced. This includes those generated by transgenic approaches, gene editing, chemical/radiation-based mutagenesis and hybridization. For transgenic lines, describe the transformation method, the number of independent lines analyzed and the generation upon which experiments were performed. For gene-edited lines, describe the editor used, the endogenous sequence targeted for editing, the targeting guide RNA sequence (if applicable) and how the editor was applied.* |
| Authentication | *Describe any authentication procedures for each seed stock used or novel genotype generated. Describe any experiments used to assess the effect of a mutation and, where applicable, how potential secondary effects (e.g. second site T-DNA insertions, mosiacism, off-target gene editing) were examined.* |

# ChIP-seq

## Data deposition

☐ Confirm that both raw and final processed data have been deposited in a public database such as GEO.

☐ Confirm that you have deposited or provided access to graph files (e.g. BED files) for the called peaks.

| | |
|---|---|
| Data access links<br>*May remain private before publication.* | *For "Initial submission" or "Revised version" documents, provide reviewer access links.  For your "Final submission" document, provide a link to the deposited data.* |
| Files in database submission | *Provide a list of all files available in the database submission.* |
| Genome browser session<br>(e.g. UCSC) | *Provide a link to an anonymized genome browser session for "Initial submission" and "Revised version" documents only, to enable peer review.  Write "no longer applicable" for "Final submission" documents.* |

## Methodology

| | |
|---|---|
| Replicates | *Describe the experimental replicates, specifying number, type and replicate agreement.* |
| Sequencing depth | *Describe the sequencing depth for each experiment, providing the total number of reads, uniquely mapped reads, length of reads and whether they were paired- or single-end.* |
| Antibodies | *Describe the antibodies used for the ChIP-seq experiments; as applicable, provide supplier name, catalog number, clone name, and lot number.* |
| Peak calling parameters | *Specify the command line program and parameters used for read mapping and peak calling, including the ChIP, control and index files used.* |
| Data quality | *Describe the methods used to ensure data quality in full detail, including how many peaks are at FDR 5% and above 5-fold enrichment.* |
| Software | *Describe the software used to collect and analyze the ChIP-seq data. For custom code that has been deposited into a community repository, provide accession details.* |

# Flow Cytometry

## Plots

Confirm that:

☐ The axis labels state the marker and fluorochrome used (e.g. CD4-FITC).

☐ The axis scales are clearly visible. Include numbers along axes only for bottom left plot of group (a 'group' is an analysis of identical markers).

☐ All plots are contour plots with outliers or pseudocolor plots.

☐ A numerical value for number of cells or percentage (with statistics) is provided.

## Methodology

| | |
|---|---|
| Sample preparation | *Describe the sample preparation, detailing the biological source of the cells and any tissue processing steps used.* |
| Instrument | *Identify the instrument used for data collection, specifying make and model number.* |
| Software | *Describe the software used to collect and analyze the flow cytometry data. For custom code that has been deposited into a community repository, provide accession details.* |

| Cell population abundance | *Describe the abundance of the relevant cell populations within post-sort fractions, providing details on the purity of the samples and how it was determined.* |
|---|---|
| Gating strategy | *Describe the gating strategy used for all relevant experiments, specifying the preliminary FSC/SSC gates of the starting cell population, indicating where boundaries between "positive" and "negative" staining cell populations are defined.* |

☐ Tick this box to confirm that a figure exemplifying the gating strategy is provided in the Supplementary Information.

# Magnetic resonance imaging

## Experimental design

| Design type | *Indicate task or resting state; event-related or block design.* |
|---|---|
| Design specifications | *Specify the number of blocks, trials or experimental units per session and/or subject, and specify the length of each trial or block (if trials are blocked) and interval between trials.* |
| Behavioral performance measures | *State number and/or type of variables recorded (e.g. correct button press, response time) and what statistics were used to establish that the subjects were performing the task as expected (e.g. mean, range, and/or standard deviation across subjects).* |

## Acquisition

| Imaging type(s) | *Specify: functional, structural, diffusion, perfusion.* |
|---|---|
| Field strength | *Specify in Tesla* |
| Sequence & imaging parameters | *Specify the pulse sequence type (gradient echo, spin echo, etc.), imaging type (EPI, spiral, etc.), field of view, matrix size, slice thickness, orientation and TE/TR/flip angle.* |
| Area of acquisition | *State whether a whole brain scan was used OR define the area of acquisition, describing how the region was determined.* |

Diffusion MRI     ☐ Used     ☐ Not used

## Preprocessing

| Preprocessing software | *Provide detail on software version and revision number and on specific parameters (model/functions, brain extraction, segmentation, smoothing kernel size, etc.).* |
|---|---|
| Normalization | *If data were normalized/standardized, describe the approach(es): specify linear or non-linear and define image types used for transformation OR indicate that data were not normalized and explain rationale for lack of normalization.* |
| Normalization template | *Describe the template used for normalization/transformation, specifying subject space or group standardized space (e.g. original Talairach, MNI305, ICBM152) OR indicate that the data were not normalized.* |
| Noise and artifact removal | *Describe your procedure(s) for artifact and structured noise removal, specifying motion parameters, tissue signals and physiological signals (heart rate, respiration).* |
| Volume censoring | *Define your software and/or method and criteria for volume censoring, and state the extent of such censoring.* |

## Statistical modeling & inference

| Model type and settings | *Specify type (mass univariate, multivariate, RSA, predictive, etc.) and describe essential details of the model at the first and second levels (e.g. fixed, random or mixed effects; drift or auto-correlation).* |
|---|---|
| Effect(s) tested | *Define precise effect in terms of the task or stimulus conditions instead of psychological concepts and indicate whether ANOVA or factorial designs were used.* |

Specify type of analysis:     ☐ Whole brain     ☐ ROI-based     ☐ Both

| Statistic type for inference<br>(See Eklund et al. 2016) | *Specify voxel-wise or cluster-wise and report all relevant parameters for cluster-wise methods.* |
|---|---|
| Correction | *Describe the type of correction and how it is obtained for multiple comparisons (e.g. FWE, FDR, permutation or Monte Carlo).* |

## Models & analysis

| n/a | Involved in the study |
|-----|------------------------|
| ☐ | ☐ Functional and/or effective connectivity |
| ☐ | ☐ Graph analysis |
| ☐ | ☐ Multivariate modeling or predictive analysis |

**Functional and/or effective connectivity**

*Report the measures of dependence used and the model details (e.g. Pearson correlation, partial correlation, mutual information).*

**Graph analysis**

*Report the dependent variable and connectivity measure, specifying weighted graph or binarized graph, subject- or group-level, and the global and/or node summaries used (e.g. clustering coefficient, efficiency, etc.).*

**Multivariate modeling and predictive analysis**

*Specify independent variables, features extraction and dimension reduction, model, training and evaluation metrics.*

