## [Peer Review File · Nature]

Manuscript Title: Predicting visual function by interpreting a neuronal wiring diagram

Reviewer Comments & Author Rebuttals

Reviewer Reports on the Initial Version:

Referees' comments:

Referee #1 (Remarks to the Author):

This manuscript details the results of an extensive effort to hand map several cell types in the optic lobe of the fly from a newly published connectome onto the visual field of the fly, then assess and hypothesize about the possible implications and functional consequence of the inferred receptive fields and interactions. The manuscript details averaged and aligned maps for two main types of fly visual neurons. Dm3 and TmY cells are interrogated and conjectured about thoroughly, as are connections to another type of cell (LC10) that forms a conduit to the central learning and association centers of the fly brain.

The work performed here done to map many cells onto the spatial map of the fly's visual system will be of great value to the field, generally, and will hopefully be published along with this work. The predictions made here are a significant contribution to fly vision and lay down some testable hypotheses that should motivate or catalyze future experiments. It is important that this kind of work comes first instead of post hoc, so that experiments can be led by theoretical insight and argument. This hones the discourse between experiment and theory, broadly, and lends some focus to experimental exploration of these networks which could take a wide (and impossible to explore exhaustively) array of different directions.

The conjectures made and consequences explored here, that follow on from the connectome of these cell types, is well-argued. However, it would be important to have some kind of hard numbers and more explanation of the methods. Hardly any parts of what is presented are quantitative. Even the visual fields lack units or concrete, quantitative connection to the other species and visual systems discussed and to which comparison is made much of in the intro and discussion. It's clear that many cells were analyzed and averaged, but how many and what the variation was like are not given. These and other concerns about the lack of qualitative results form a major concern. Given this, it is impossible to assess the validity of the statistical treatment of these data and results.

The conclusions drawn in the work are largely impressions and qualitative hypotheses. These are at times well-reasoned, but at other points in the manuscript rather tenuous. This is fine as an overall approach to these data - to comb through, extract a pattern that could have meaningful functional consequences and then - propose it - but the synthesis lacks some power and depth.

Several suggestions made below are offered as constructive advice for improving the manuscript. There are many places where more detail would be useful, and where a reference of explanation could be added. Overall, the manuscript suffers from being a rather telegraphic report that runs through a large amount of different cell types and connections without much depth in any one section. That combined with the acronym-heavy text might make this more readable by a fly-visual-system specialist audience.

Suggestions:

- 1) The results text could use more logical or narrative links between sections.
- 2) The “illusory responses” promise in the abstract is not sufficiently explained in the intro or results and should be shored up with some quantitative metrics on the size of the proposed effect and some way to compare it to what is observed in mammals. This would help non-fly-expert readers, especially.
- 3) Three orientations doesn’t at all have to imply a coarse orientation representation, that depends on the fidelity of the readout of those three directions. Quantitative comparisons would help here, and some context for this claim should be given. Otherwise, this feels like an empty and potentially misleading statement.
- 4) A compelling summary cartoon would be nice to ground the reader before launching into the fly brain and individual neuronal traces. It would be good to see an illustration of the illusory perception predicted in flies, with appropriate visual angles included, and a connection to known fly location sensing discrimination limits.
- 5) Is Figure 5h known or predicted by this work (or the recent fly connectome)?
- 6) Input output plots are a bit unilluminating - is there some way to synthesize this a bit better for the reader?
- 7) The main result is the manual assignment of Dm3 and TmY cells to a spatial lattice - is there any notion of possible errors in this procedure? Any details on how it was validated? Any details on inter-individual variation in flies (since this is from one female fly only)? Will the manually mapped data be available? How many cells were hand mapped? How many of each type? Any kind of error bars on this?
- 8) Fig 3e-f make a major point of the paper but are a little hard to parse, and the functional consequence of this connectivity isn’t clear. Can this be, broadly, improved in some way, perhaps in connection/conjunction with the answer to the previous point?
- 9) It would be nice to have a summary of the punchline(s) of each figure summarized in a sentence to lead off the captions, since the paper, overall, is very raw.
- 10) How strong is the summation assumption for the CRFs? Was any consideration made of gap junction connections, that are all over the fly brain and implicated in computation?
- 11) Why should we think the inter type connections are inhibitory except for the appeal to a parallel with mammalian vision? What if they aren’t?
- 5) Is there enough evidence for the LC10e junction detector claim? It seems tenuous. Some quantification might help this, some model perhaps?

Referee #2 (Remarks to the Author):

1. Summary of the key results

The current work attempts to interpret local visual processing for edge detection from a subpart of a visual wiring diagram of the fruit fly. The author identifies a circuit composed of eight cell types (3 TmY, 3 DM3, and LC15, LC10e), whose connectivity the author speculates to support functional edge extraction.

Specifically, the observation that all the cells TmY and DM3 show dendrites extending in specific directions suggests that these cells respond selectively to oriented visual stimuli along three main orientations reflecting the hexagonal lattice of the ommatidia. Furthermore, inhibitory connections between cells with different orientation tuning suggest a sharpening of orientation detection. Also, the excitation between neighbouring cells with the same orientation tuning suggests a neural substrate for the perception of illusory contours. LC10e and LC15, by summing activity of cells of different orientation tuning may extract corners or respond to the presence of any orientation.

These motifs echo some of the ones hypothesised for mammalian visual cortex.

2. Originality and significance: if not novel, please include reference

The prediction of functional circuits from the interpretation of wiring diagrams is a rather recent approach in insects, which is here possible thanks to the recent connectome of a visual system, uniquely available in *Drosophila*. Similar analyses of *Drosophila* connectomes, attempting to infer potential computational function from the circuit, have already been achieved regarding other brain areas such as the central complex (Hulse et al., 2021) or the mushroom bodies (Eschbach et al., 2020).

As acknowledged by the author, the supposed functions achieved by the identified motifs are already known. Extraction of edges along these three directions has already been proposed and demonstrated behaviourally (Srinivasan et al., 1994) or electrophysiologically (e.g., Stadele et al. 2020). The perception of illusory contours has been reported in other insects (recently reviewed in (Gatto et al., 2022)), and thus is likely to exist in flies. And the similarity of early visual processing in insects and vertebrates has already been highlighted (Sane and Zipursky 2010).

Altogether, conceptually, this paper does not provide any novel or significant advance. However, its contribution is to identify an actual neural substrate that may support these various visual computations, and thus provide neural predictions that could henceforth be tested thanks to the neurogenetic tools available in flies.

Refs :

Hulse, Brad K., et al. "A connectome of the *Drosophila* central complex reveals network motifs suitable for flexible navigation and context-dependent action selection." *Elife* 10 (2021).

Eschbach, Claire, et al. "Recurrent architecture for adaptive regulation of learning in the insect brain."

Nature Neuroscience 23.4 (2020): 544-555.

Srinivasan, M. V., Zhang, S. W., & Witney, K. (1994). Visual discrimination of pattern orientation by honeybees: performance and implications for 'cortical' processing. *Philosophical Transactions of the Royal Society of London. Series B: Biological Sciences*, 343(1304), 199-210.

Sanes, Joshua R., and S. Lawrence Zipursky. "Design principles of insect and vertebrate visual systems." *Neuron* 66.1 (2010): 15-36.

3. Data & methodology: validity of approach, quality of data, quality of presentation

Overall, the figures are of good quality. They show neurons and their reconstructed visual field/synaptic maps, which can be understood intuitively. However, the motifs presented sometimes lack proper quantification (e.g., Fig S6 'the maps look more noisy', [...] 'the maps often resemble two edges in a corner configuration'). Also, there is a complete lack of statistics. Notably, average synaptic counts are presented for the input maps, but how they vary across cells, or how many neurons are used for averaging is unclear.

4. Conclusions: robustness, validity, reliability

Inferring functions from connectomes will always remain speculative, but the aim of this approach is to provide new insights about the function of neural circuits together with testable predictions. In this respect the hypotheses advanced by the author seem robust, all the more because they point at functions that are congruent with functions we already know exist in insects. Their validity remains to be tested.

5. Suggested improvements: experiments, data for possible revision

We do not have specific suggestions for improvement beyond the limitations already highlighted by the author:

'One might criticize this paper as incomplete, because its predictions are crying out to be tested by neurophysiology.' [...] 'One might also complain that this paper is incomplete without modeling the dynamics of neural activity based on the neuronal wiring diagram'

6. References: appropriate credit to previous work?

The references are appropriate. Adding the following could further improve the manuscript.

Gatto et al., 2022, recently reviews the work achieved in the perception of illusory contours in insects.

Gatto, E., Loukola, O. J., Petrazzini, M. E. M., Agrillo, C., & Cutini, S. (2022). Illusional Perspective across Humans and Bees. *Vision*, 6(2), 28.

The work of Sane and Zipursky (2010) reviewing the common design principles of insect and vertebrate visual system should be mentioned.

Sanes, Joshua R., and S. Lawrence Zipursky. "Design principles of insect and vertebrate visual systems." *Neuron* 66.1 (2010): 15-36.

The statement 'T2a is predicted to be a transient ON-OFF cell' lacks a reference. Perhaps citing (Keleş et al., 2020), which characterised T2 and T3 responses to be ON-OFF would be appropriate to support such claim.

Keleş, Mehmet F., et al. "Inhibitory interactions and columnar inputs to an object motion detector in *Drosophila*." *Cell reports* 30.7 (2020): 2115-2124.

7. Clarity and context: lucidity of abstract/summary, appropriateness of abstract, introduction and conclusions

The manuscript could be improved. Notably, in the current introduction, it is not obvious what question is being addressed. The six last paragraphs of the introduction are very specific of the cells targeted, and could rather be treated directly in the result section. The introduction could thus be largely shortened to highlight instead what general problems are being addressed, how they are addressed, and why these are significant.

The discussion also feels like a continuation of the results section. It is presented as a list of independent and very specific sections, and maintain a speculative tone all along. These speculations will certainly interest specialists of the topic. However, it lacks a more general and comprehensive synthesis to highlight the significance of the neural motif identified to the general reader.

We hope our review was helpful,

Referee #3 (Remarks to the Author):

Overview:

Seung provides a connectomic analysis for a set of connected neurons from the optic lobe of *Drosophila*, using connectomic electron microscope data from a single, female fly brain. Seung describes inputs and outputs of two major cell types, Dm3 and TmY, their subtypes, and the connectivity between them.

Seung then uses the fact that their input neurons have an approximately one-to-one correspondence with single ommatidia in order to illustrate predictions for their functional receptive fields. Seung further describes downstream neurons, LC10e and LC15, that integrate inputs from these cell types. Again, Seung uses their inputs from Dm3 and TmY to make predictions about their functional receptive fields. The work is well written and easy to follow. Because there is little physiological data for these cell types, the connectomics approach taken here means the reader is left with several unanswered questions about the functional properties of these cells, and their role within the visual system. Nevertheless, this work provides a large volume of useful data in the process towards that goal. While the addition of functional analyses, whether from physiological recordings or from computational modelling, would make this work an exceptional feat of progress, it would also make the manuscript enormous and more difficult to parse. As it stands, it provides helpful predictions that can be used as hypotheses for future studies.

Main concerns:

The main concern I have is with the use of the term “classical receptive field (CRF)”. Although Seung does, at points, make it clear that this work cannot describe actual CRFs – which are determined by a neuron’s functional response to the stimulus space – the term “CRF” is often used as though real CRFs are being reported in the results. I recommend using a variant of that term, e.g. pCRF (predicted CRF), or something similar, to avoid any confusion amongst less experienced readers who are less familiar with this nuance. I don’t think it is necessary to use pCRF (or equivalent) for the phrase “beyond the CRF”, where Seung describes modulatory effects on the CRF, whatever the actual CRF is.

Minor concerns:

1) The Introduction is easy to follow, but reads more like a Discussion, laying out all the results that haven’t yet been presented, and relating them to previously published work. I think the Introduction would greatly benefit from a description of more background literature that motivates the scientific work undertaken in this paper and the cells that have been chosen for analyses, before referring to the results.

2) Introduction>par. 3:

The works of Hubel and Wiesel would be good to cite in this paragraph, as some of the earliest evidence for orientation selective receptive fields.

3) Results>Dm3 receives input from collinear columns>par. 6:

Looks more like Dm3 pCRFs are 5x1, or even larger. Please describe in the text, or in the Methods, where the number 3 comes from.

4) Results>Dm3 receptive field estimates>par. 4:

I find this paragraph difficult to determine which citation is responsible for which physiological property that is being reported. Are all relevant citations present here?

5) Results>Cross-orientation inhibition>par. 5:

Citation is needed for the sentence "These neurotransmitters are typically inhibitory in the fly brain, so Dm3 will be assumed inhibitory".

6) Results>LC10e pathways to the central complex:

This section appears to be purely discussion, rather than presenting any new results. Was there meant to be a reference to a figure here? If not, then I think it is best moved to the Discussion.

7) Results>ON-OFF input regions>par. 2 and par. 4:

Please show data for all reported results. Not only is this more transparent, but such data might also be useful for the reader.

8) Results>ON-OFF input regions>par. 4:

Is Fig. 6f mean to read 6b?

9) Discussion:

I think it would be very helpful to list or tabulate the main predictions near the beginning of the Discussion, especially for those who have the capacity to verify the predictions in future work.

10) Discussion>par.3>"are now empirical facts in the fly optic lobe":

I think stating this as fact is going too far, given that excitation and inhibition can only be verified by appropriate functional studies.

11) Discussion>OFF input regions as receptive field predictions>par. 1>"Orientation selective (OS) neurons were predicted to span a maximum of.....":

It seems as though the results presented in this study actually contradict the predictions of Horridge (2003), as they are all larger by at least one ommatidium. However, these predictions were made for the honeybee, so do not have to be supported accurately by the results here. It would be helpful to state this. Can the reasoning behind Horridge's predictions be applied to *Drosophila* to make new, *Drosophila*-specific predictions? That would be more informative than whether or not Horridge's predictions have been confirmed, per se, which doesn't further our scientific understanding of the results.

12) Methods>Spatial coordinates in the optic lobe>par. 1>"These rules are simplified:...":

How is this simplification being used in the current study? As it is, this sentence confuses me, as the main text refers to the number of putative synapses as a measure of connectivity.

13) Methods>Spatial coordinates in the optic lobe>par. 2>"I manually assigned all Mi1 cells to locations in a hexagonal lattice":

What considerations were made when doing this assignment manually? What caveats to this method are there, e.g. I suspect that this becomes a particularly unhelpful representation towards the periphery of the eye, or anywhere there is high curvature on the surface of the eye. Is this performed locally for each cell that was analysed, as I suspect a hexagonal lattice breaks down beyond a certain distance across the whole eye, e.g. because of dislocations in the lattice.

14) Fig. 1 and other similar panels in all figures:

I strongly recommend adding axes to help the reader properly orient these diagrams.

15) Figs. 1, 2, 5 and 6:

There are some long numbers in the captions that don't seem to represent anything, presumably a formatting error.

16) Fig. S2:

It looks as though, when grouped in panel D, TmY4/TmY9 receive more Tm1 inputs than the sum of TmY9qL and TmY9q in panel B, with TmY4 not even showing on panel B.

Response to reviews of original submission

I'm grateful to the Reviewers for their careful reading of the original submission. Stimulated by their suggestions, I have prepared a major revision that uses the concept of the extraclassical receptive field (ERF) to make predictions more concrete and specific, adds more quantification through various summary statistics, gives a complete picture of cell-to-cell variation in the Supplementary Data, and describes the Methods in detail.

The major changes are:

1. Extraclassical receptive fields (ERFs) are now explicitly predicted (new Figs. 4, 5, Data S3, S4) by mapping disynaptic pathways from Tm1 to Dm3 and Tm1 to TmY. The ERF predictions are testable by visual physiologists. The ERFs predicted for TmY clarify the implications for completion of noisy or illusory contours.
2. The original manuscript (old Figs. 1 and 2, last panels) approximated the CRF predictions for Dm3 and TmY with ellipses, which effectively served as descriptive statistics for the CRF predictions. Since the reviewers were not satisfied by visual display of the ellipses, boxplots have been added for the ellipse parameters (orientation, aspect ratio).
3. 1D projections of the receptive fields have been added, to provide further descriptive statistics that complement the ellipses.
4. Ellipse approximations are also provided for the predicted ERFs (Figs. 4, 5), and the predicted receptive fields of LC10e and LC15 as well (Fig. 6).
5. To complement the static renderings of cells in the main Figures, Data S1 provides interactive 3D visualizations through Neuroglancer links.
6. Data S2 provides CSV files of the coordinates of the relevant hexel types on the hexagonal lattice, and SVG files with renderings of Mi1 and Tm1 that show the quality of the spatial mapping.
7. Data S3, S4, and S5 supply connectivity maps for 2000+ individual Dm3, TmY, and LC cells (1273, 572, and 77 cells). This complete description of cell-to-cell variability complements the averages and summaries in the main figures. All Supplementary Data is downloadable from Zenodo or Google Drive through links provided in the revised manuscript.
8. The original Figs. 3 and 4 (spatial configuration of connectivity between Dm3 and TmY cells) have been moved to Fig. S6 in the revised manuscript.
9. The Methods section has been greatly expanded.
10. All analyses have been "upgraded" to v783 of the proofreading. The original submission was based on v630.
11. The title has been changed to emphasize "prediction" rather than "insight."

Reviewer 1

This manuscript details the results of an extensive effort to hand map several cell types in the optic lobe of the fly from a newly published connectome onto the visual field of the fly, then assess and hypothesize

about the possible implications and functional consequence of the inferred receptive fields and interactions. The manuscript details averaged and aligned maps for two main types of fly visual neurons. Dm3 and TmY cells are interrogated and conjectured about thoroughly, as are connections to another type of cell (LC10) that forms a conduit to the central learning and association centers of the fly brain.

The work performed here done to map many cells onto the spatial map of the fly's visual system will be of great value to the field, generally, and will hopefully be published along with this work.

FlyWire is committed to making all data publicly available. Since the original submission, maps of 31 cell types have been made publicly available at the FlyWire Codex, and a technical report about the maps is in preparation.

https://codex.flywire.ai/app/visual_columns_map

Data S2 provides mappings of hexel cell types to a hexagonal lattice, along with visualizations.

A "hexel cell type" is defined as a cell type that is in one-to-one correspondence with ommatidia and has a receptive field center that is roughly one ommatidium wide (Methods).

The predictions made here are a significant contribution to fly vision and lay down some testable hypotheses that should motivate or catalyze future experiments. It is important that this kind of work comes first instead of post hoc, so that experiments can be led by theoretical insight and argument. This hones the discourse between experiment and theory, broadly, and lends some focus to experimental exploration of these networks which could take a wide (and impossible to explore exhaustively) array of different directions.

Thanks for this insightful description of the relationship between experiment and theory. I have incorporated these thoughts into the revised abstract and introduction.

The conjectures made and consequences explored here, that follow on from the connectome of these cell types, is well-argued. However, it would be important to have some kind of hard numbers and more explanation of the methods. Hardly any parts of what is presented are quantitative. Even the visual fields lack units or concrete, quantitative connection to the other species and visual systems discussed and to which comparison is made much of in the intro and discussion. It's clear that many cells were analyzed and averaged, but how many and what the variation was like are not given.

The numbers of Dm3 and TmY cells were given in the text of the original manuscript. They have now been moved to Fig. 1 and 2 captions, to make them more noticeable. I apologize for previously omitting the numbers of LC10e and LC15 cells; these have now been added to the text and Fig. 6 caption. The numbers were implicit in the original manuscript, which showed the connectivity maps for all LC10e and LC15 cells in Figs. S6 and S7.

The original Figs. 1g and 2k summarized cell-to-cell variation for Tm1-Dm3 and Tm1-TmY maps by drawing ellipses. This method of characterizing variation is now extended to maps of disynaptic pathways leading to Dm3, TmY, and LC cells (Figs. 4, 5, and 6). Summary statistics of the ellipses are given in boxplots.

To complement the ellipses, 1D projections of predicted receptive fields are also computed and shown.

Finally, Data S3, S4, and S5 provide maps for all 2000+ Dm3, TmY, and LC cells to give a complete picture of variation.

Spatial coordinates rely on a single unit, the lattice constant for the hexagonal grid. The visual angle subtended by a lattice constant varies with location in the eye, as characterized by (Zhao et al. 2022).

The Discussion now includes the caveat that visual acuity and lattice spacing differ substantially between *Drosophila* and *Apis*, along with references.

All heatmaps are normalized. The maximum values for colormaps were given in the figure legends in the original manuscript. The maximum values are now usually printed in the figure itself, so that they will not be missed.

These and other concerns about the lack of qualitative results form a major concern. Given this, it is impossible to assess the validity of the statistical treatment of these data and results.

I'd like to clarify that I am providing descriptive statistics for the generation of hypotheses. There are no *p*-values as I am not testing hypotheses.

The conclusions drawn in the work are largely impressions and qualitative hypotheses. These are at times well-reasoned, but at other points in the manuscript rather tenuous. This is fine as an overall approach to these data - to comb through, extract a pattern that could have meaningful functional consequences and then propose it - but the synthesis lacks some power and depth.

Several suggestions made below are offered as constructive advice for improving the manuscript. There are many places where more detail would be useful, and where a reference of explanation could be added. Overall, the manuscript suffers from being a rather telegraphic report that runs through a large amount of different cell types and connections without much depth in any one section. That combined with the acronym-heavy text might make this more readable by a fly-visual-system specialist audience.

Suggestions:

- 1) The results text could use more logical or narrative links between sections.

I hope that the logic is now clearer, while respecting the constraint of brevity.

2) The “illusory responses” promise in the abstract is not sufficiently explained in the intro or results and should be shored up with some quantitative metrics on the size of the proposed effect and some way to compare it to what is observed in mammals. This would help non-fly-expert readers, especially.

The original manuscript relied on abstract cartoons (Figs. 3e, 3f, 4e, 4f) to predict consequences of connections between Dm3 and TmY cells. The cartoons have now been replaced by concrete predictions of extraclassical receptive fields (ERFs). These are a well-known kind of measurement familiar to visual neurophysiologists. Since the TmY ERFs make “collinear facilitation” concrete, the implications for the completion of noisy or illusory contours should be more obvious.

It is asking too much from connectomics to predict the size of effects. Neurophysiology is needed. Synapse count gives us limited information about physiological strength of a connection, but many parameters are currently unknown. For example (Ammer et al. 2023) have argued that the unitary conductance of inhibitory synapses can be at least an order of magnitude larger than that of excitatory synapses.

3) Three orientations doesn't at all have to imply a coarse orientation representation, that depends on the fidelity of the readout of those three directions. Quantitative comparisons would help here, and some context for this claim should be given. Otherwise, this feels like an empty and potentially misleading statement.

The sentence has been removed.

4) A compelling summary cartoon would be nice to ground the reader before launching into the fly brain and individual neuronal traces. It would be good to see an illustration of the illusory perception predicted in flies, with appropriate visual angles included, and a connection to known fly location sensing discrimination limits.

I have added a summary cartoon (Fig. 1a) to show neuropils of the optic lobe and the major cell types.

I have also added a cartoon (Fig. 3e) to define the end zones and side zones of the CRF.

5) Is Figure 5h known or predicted by this work (or the recent fly connectome)?

In principle, all the relevant cell types and connections reside in the data released by (Dorkenwald et al. 2023; Matsliah et al. 2023; Schlegel et al. 2023). However, some mining of these datasets is required to expose the information. The Dm3 and TmY types are barely mentioned in (Matsliah et al. 2023), which sends the reader to the preprint of the present work for more information.

Therefore, the present paper can be regarded as reporting the discovery of the third Dm3 type and the splitting of TmY9 into two types. It can also be regarded as the first report of the connectivity of these types. In effect, Fig. 5h (now Fig. 3b) is presenting connectivity patterns discovered by the present work. These patterns are not predictions though. My predictions pertain to physiology, not anatomy.

6) Input output plots are a bit unilluminating - is there some way to synthesize this a bit better for the reader?

The reviewer is presumably concerned with Figs. S1, S2, S4, and S5. These figures are intended to quantify the importance of the kinds of connectivity studied in the main text. But they could turn out to be useful to researchers for other reasons. In this respect, the present work serves as an information resource that is a deep dive into the Dm3-TmY circuit. It is complementary to (Matsliah et al. 2023), a shallow and broad survey of all cell types intrinsic to the optic lobe.

7) The main result is the manual assignment of Dm3 and TmY cells to a spatial lattice

As explained in the Methods, Mi1 is the only cell type that was manually mapped. Mi1 locations were automatically propagated to other hexel types (Tm1, Tm2, etc) by applying the Hungarian algorithm to the connectivity matrix.

Dm3 and TmY spatial coordinates were automatically computed by processing the Tm1-Dm3 and Tm1-TmY connectivity maps.

- is there any notion of possible errors in this procedure?

For any given hexel type, a cell may be lacking for up to 10% of lattice points. This is partially due to underrecovery of cells in v783 proofreading, which is discussed in the Methods of (Matsliah et al. 2023). It is also due to underrecovery of connections by automated synapse detection.

Any details on how it was validated?

Coordinate assignments were primarily validated by visualizations, which are now provided in Data S2.

Any details on inter-individual variation in flies (since this is from one female fly only)?

This analysis has been done for one optic lobe of a single fly.

Will the manually mapped data be available?

The maps for hexel types are now included in Data S2.

Since the original submission, maps of 31 cell types have been made publicly available at the FlyWire Codex.

https://codex.flywire.ai/app/visual_columns_map

In general, FlyWire is committed to making all data publicly available.

How many cells were hand mapped?

Almost 800 Mi1 cells were manually assigned to a hexagonal grid.

How many of each type?

All other types were assigned automatically, and the resulting numbers are given in the Methods.

Any kind of error bars on this?

The visualizations in Data S2 show that the coordinate assignments are clean. If they were not clean, the predicted CRFs in Figs. 1 and 2 could not look so good.

8) Fig 3e-f make a major point of the paper but are a little hard to parse, and the functional consequence of this connectivity isn't clear. Can this be, broadly, improved in some way, perhaps in connection/conjunction with the answer to the previous point?

I'm grateful for this suggestion, which motivated a major revision and improvement of the manuscript.

The cartoons in old Figs. 3e-f and 4e-f have been replaced by the ERF predictions of new Figs. 4 and 5. The old Figs. 3c-d and 4c-d about connectivity between Dm3 and TmY types have been moved to new Fig. S6.

9) It would be nice to have a summary of the punchline(s) of each figure summarized in a sentence to lead off the captions, since the paper, overall, is very raw.

The titles of the figure legends have been lengthened slightly to make them more informative.

10) How strong is the summation assumption for the CRFs?

Because the synapses that contribute to "CRF"s are close together, the neuron is likely to be equipotential for those synapses. Therefore the synaptic conductances are expected to influence the neuron via their sum. The synapses that contribute to "ERF"s can be distant from each other and from the synapses that contribute to "CRF"s, so it is less clear whether linear summation holds when "ERF" synapses are included.

The above statements concern summation of the *outputs* of presynaptic cells. If we consider the inputs to the presynaptic cells, or the visual stimulus, then nonlinearities will of course intervene. An obvious example is threshold nonlinearity in the presynaptic cell. That being said, receptive fields of hexel cell types have been characterized using white noise stimuli (Arenz et al. 2017; Drews et al. 2020), so linearity is evidently a reasonable approximation around some operating point.

Was any consideration made of gap junction connections, that are all over the fly brain and implicated in computation?

Unfortunately the connectomic data does not yet contain gap junctions.

11) Why should we think the inter type connections are inhibitory except for the appeal to a parallel with mammalian vision? What if they aren't?

The evidence and logic are now explained in detail in the Methods. Briefly, Dm3 is predicted glutamatergic from transcriptomic data (Davis et al. 2020), and from EM images (Eckstein et al. 2020). Glutamate is often inhibitory (Liu and Wilson 2013), but this depends on the identity of the glutamate receptor in the postsynaptic neuron. Dm3 (Davis et al. 2020; Kurmangaliyev et al. 2020), TmY4 and TmY9 (Y. Kurmangaliyev, personal communication) express the GluCl α receptor at relatively high levels, so Dm3 is presumed inhibitory in its effects on Dm3 and TmY targets.

That being said, existing transcriptomic studies do not have subcellular resolution, so it is not known whether the GluCl α receptor is located at synapses from Dm3. If it turns out that Dm3 is excitatory, that would be an interesting surprise. I cannot guarantee that the ideas in the paper are correct, but they are plausible enough that they will be interesting even if they are wrong.

5) Is there enough evidence for the LC10e junction detector claim? It seems tenuous. Some quantification might help this, some model perhaps?

I have made two improvements. First, Figure 6 now contains ellipse summaries of LC10e inputs, which are evidence for a systematic spatial relationship between the presynaptic orientation detectors, although there is also considerable noise. Second, I have tried to make the assumptions underlying the conjecture more explicit.

Reviewer 2

1. Summary of the key results

The current work attempts to interpret local visual processing for edge detection from a subpart of a visual wiring diagram of the fruit fly. The author identifies a circuit composed of eight cell types (3 TmY, 3 DM3, and LC15, LC10e), whose connectivity the author speculates to support functional edge extraction.

Specifically, the observation that all the cells TmY and DM3 show dendrites extending in specific directions suggests that these cells respond selectively to oriented visual stimuli along three main orientations reflecting the hexagonal lattice of the ommatidia. Furthermore, inhibitory connections between cells with different orientation tuning suggest a sharpening of orientation detection. Also, the excitation between neighbouring cells with the same orientation tuning suggests a neural substrate for the perception of illusory contours. LC10e and LC15, by summing activity of cells of different orientation tuning may extract corners or respond to the presence of any orientation.

These motifs echo some of the ones hypothesised for mammalian visual cortex.

2. Originality and significance: if not novel, please include reference

The prediction of functional circuits from the interpretation of wiring diagrams is a rather recent approach in insects, which is here possible thanks to the recent connectome of a visual system, uniquely available in *Drosophila*. Similar analyses of *Drosophila* connectomes, attempting to infer potential computational function from the circuit, have already been achieved regarding other brain areas such as the central complex (Hulse et al., 2021) or the mushroom bodies (Eschbach et al., 2020).

I admire the cited works for their advances in linking structure with function. The central complex is of course a stunning example of this, and I often mention (Hulse et al. 2021) as a landmark achievement of HHMI Janelia. But historically the discovery of compass neurons was made by physiologists, and later on connectomic analyses attempted to explain the mechanisms underlying what the physiologists had discovered. One can imagine a counterfactual history in which (Hulse et al. 2021) predicted compass neurons by analyzing the central complex wiring diagram, and physiologists later confirmed the prediction. But that's not how history unfolded.

Similarly, physiologists already knew that DANs carry reward/punishment/teaching signals, and MBONs signal the learned valences of stimuli. The connectome helps one figure out the mechanisms supporting those functions (Eschbach et al. 2020).

The novelty of the present work is that I am predicting the functions of all cell types in a novel circuit BEFORE the physiologists have done their experiments. This step might seem obvious and natural to the reviewer, but everyone I've met seems surprised by it.

I thank the reviewer for these citations, which are included for contrast in the revised manuscript.

As acknowledged by the author, the supposed functions achieved by the identified motifs are already known. Extraction of edges along these three directions has already been proposed and demonstrated behaviourally (Srinivasan et al., 1994) or electrophysiologically (e.g., Städele et al. 2020).

It is difficult or impossible to discover neural mechanisms based on behavioral experiments alone. If it were so easy, there would be no need for neuroscience.

There is no electrophysiological demonstration of "Extraction of edges along these three directions" in (Städele et al. 2020). They report that LC15 responds to bars of any orientation, as cited in my original manuscript. This is a report of LACK OF orientation tuning.

The perception of illusory contours has been reported in other insects (recently reviewed in (Gatto et al., 2022)), and thus is likely to exist in flies.

(Gatto et al. 2022) is unfortunately misleading concerning this issue. Their review contains no citations to insect perception of illusory contours other than the old reports of Adrian Horridge and colleagues in bees. As I explained in my original manuscript, Horridge himself has retracted his old claim that bees perceive illusory contours. For convenience, I reproduce the relevant passage from his book (Horridge 2009):

By 1993, it was possible to 'suggest, perhaps for the first time, the existence of feature-extracting mechanisms in the insect visual system that might be comparable, functionally, to those known to exist in the mammalian cortex' (Srinivasan et al. 1993). This lyric was inspired by an inference that insects perceived illusory contours. When they had been trained to discriminate between the orientations of shuffled orthogonal gratings, bees apparently saw the contours of the Kanizsa rectangle illusion (van Hateren et al. 1990:Fig. 4). It was supposed that, as in the human cortex, lines of edge detectors with similar orientation were strung together. Bees, like humans, also responded as though they saw an illusory orientation at a fault line across a regularly striped pattern (Horridge et al. 1992). There seemed to be nothing wrong with the idea of illusory contours, but at the time we did not know that different edge orientations in close proximity cancelled each other or that edge detectors did not span across gaps that were resolved.

When the experiments were repeated, they failed. There had been two changes to the design of the experiments. Until 1996, there were no baffles in the apparatus so the bees could enter at full speed and make a fast decision from further away. Also, they were allowed 10 minutes and two visits on each side in the tests, which allowed them to improve their success rate. After 1996, however, the baffles halted them and they took longer to peruse the targets from a fixed distance. They also had only five minutes on each side and many varied tests were intercalated, so they saw the same test at long intervals between other tests. With these precautions, the bees did not detect illusory edges or the edges at fault lines (Horridge 2003a). Also, David O'Carroll told me that he could not repeat the detection of illusory contours when recording from single neurons of the dragonfly lobula.

So contrary to (Gatto et al. 2022), Horridge came to the conclusion that bees do NOT see illusory contours, because his hypothetical edge detectors do not work that way.

As Horridge makes clear in his book, figuring out what insects can perceive purely by behavioral experiments has been highly challenging. The present work suggests a new approach to studying illusory contours starting with neurophysiology. First the CRFs and ERFs of Dm3 and TmY cells should be measured and compared with my predictions. The findings should be used to constrain a neural network model based on the connectivity patterns described by this paper. Then one should search for a visual stimulus that excites a TmY cell in the model even without stimulus contrast inside the TmY CRF. Then this visual stimulus should be applied to a real fly, and TmY responses should be measured. If this stimulus results in neurophysiological evidence of illusory contour responses, then one can proceed to use the same stimulus in behavioral experiments.

And the similarity of early visual processing in insects and vertebrates has already been highlighted (Sane and Zipursky 2010).

Parallels have generally been drawn with vertebrate retina, and this reference is no exception. The present work is novel because it points out possible parallels with mammalian cortex.

Altogether, conceptually, this paper does not provide any novel or significant advance. However, its contribution is to identify an actual neural substrate that may support these various visual computations, and thus provide neural predictions that could henceforth be tested thanks to the neurogenetic tools available in flies.

I am delighted by the reviewer's summary of the contribution. Indeed I will be happy if the present work can inspire physiology experiments. Regarding novelty, many people have been surprised that structural analysis can make such detailed predictions about visual responses of a neural circuit, when none of the neurons in that circuit have ever been studied by physiologists.

Refs :

Hulse, Brad K., et al. "A connectome of the Drosophila central complex reveals network motifs suitable for flexible navigation and context-dependent action selection." *Elife* 10 (2021).

Eschbach, Claire, et al. "Recurrent architecture for adaptive regulation of learning in the insect brain." *Nature Neuroscience* 23.4 (2020): 544-555.

Srinivasan, M. V., Zhang, S. W., & Witney, K. (1994). Visual discrimination of pattern orientation by honeybees: performance and implications for 'cortical' processing. *Philosophical Transactions of the Royal Society of London. Series B: Biological Sciences*, 343(1304), 199-210.

Sanes, Joshua R., and S. Lawrence Zipursky. "Design principles of insect and vertebrate visual systems." *Neuron* 66.1 (2010): 15-36.

3. Data & methodology: validity of approach, quality of data, quality of presentation

Overall, the figures are of good quality. They show neurons and their reconstructed visual field/synaptic maps, which can be understood intuitively.

However, the motifs presented sometimes lack proper quantification (e.g., Fig S6 'the maps look more noisy', [...] 'the maps often resemble two edges in a corner configuration').

More summary statistics are now displayed for quantification. Ellipse approximations, previously supplied only for “CRF”s, are now provided for maps of all cell types. Ellipse parameters are displayed in boxplots. 1D projections of “CRF”s and “ERF”s provide nonparametric measures of “receptive field” properties. Displacement vectors are plotted to support the claim of spatial relations between the orientation detectors presynaptic to LC10e.

Average “ERF”s are now systematically ranked by estimated strength to select a subset for display. “CRF”s and “ERF”s for individual cells are provided in Data S3-S5 to provide further insights into variability across cells.

Also, there is a complete lack of statistics. Notably, average synaptic counts are presented for the input maps, but how they vary across cells, or how many neurons are used for averaging is unclear.

In the original manuscript, the numbers of Dm3 and TmY cells were provided in the text. In the revised submission, the numbers have been moved to the figure captions, in the hope that they may be more noticeable there. The original manuscript showed ellipse approximations to CRFs to illustrate variability across cells. Summary statistics about the ellipses are now shown in boxplots. Longitudinal and transverse projections of RFs are also provided along with standard deviations as further descriptive statistics to complement the ellipses.

4. Conclusions: robustness, validity, reliability

Inferring functions from connectomes will always remain speculative, but the aim of this approach is to provide new insights about the function of neural circuits together with testable predictions. In this respect the hypotheses advanced by the author seem robust, all the more because they point at functions that are congruent with functions we already know exist in insects. Their validity remains to be tested.

5. Suggested improvements: experiments, data for possible revision

We do not have specific suggestions for improvement beyond the limitations already highlighted by the author:

‘One might criticize this paper as incomplete, because its predictions are crying out to be tested by neurophysiology.’ [...] ‘One might also complain that this paper is incomplete without modeling the dynamics of neural activity based on the neuronal wiring diagram’

6. References: appropriate credit to previous work?

The references are appropriate. Adding the following could further improve the manuscript.

Gatto et al., 2022, recently reviews the work achieved in the perception of illusory contours in insects.

Gatto, E., Loukola, O. J., Petrazzini, M. E. M., Agrillo, C., & Cutini, S. (2022). Illusional Perspective across Humans and Bees. *Vision*, 6(2), 28.

I have now cited (Gatto et al. 2022) to clear up possible misconceptions concerning illusory contours. Judging from this review, researchers seem unaware that (Horridge 2009) has renounced his old claims of illusory contour perception by bees. (Gatto et al. 2022) gives the misleading impression that illusory contour perception is well-established in insects. In fact, there are no such claims in the literature, aside from the now-disowned claims of Horridge and colleagues.

The work of Sane and Zipursky (2010) reviewing the common design principles of insect and vertebrate visual system should be mentioned.

Sanes, Joshua R., and S. Lawrence Zipursky. "Design principles of insect and vertebrate visual systems." *Neuron* 66.1 (2010): 15-36.

(Sanes and Zipursky 2010) is beautifully written but concerns development, which is extremely interesting but not the focus of the present work. Therefore I hope that the reviewer will not mind if I instead cite the more recent (Borst and Helmstaedter 2015) and (Clark and Demb 2016) concerning parallel relationships between wiring and function. I emphasize again that all of these reviews focus on comparing fly with vertebrate retina, not cortex.

The statement 'T2a is predicted to be a transient ON-OFF cell' lacks a reference. Perhaps citing (Keleş et al., 2020), which characterised T2 and T3 responses to be ON-OFF would be appropriate to support such claim.

Keleş, Mehmet F., et al. "Inhibitory interactions and columnar inputs to an object motion detector in *Drosophila*." *Cell reports* 30.7 (2020): 2115-2124.

I have clarified that this claim is a prediction based on T2a input connectivity. It is a prediction of the present work, not an experimental finding from another paper.

Following the suggestion of the reviewer, I have also added the citation to (Keleş et al. 2020) for comparison because T3 (and to a lesser extent T2) have connectivity similar to T2a.

7. Clarity and context: lucidity of abstract/summary, appropriateness of abstract, introduction and conclusions

The manuscript could be improved. Notably, in the current introduction, it is not obvious what question is being addressed. The six last paragraphs of the introduction are very specific of the cells targeted, and could rather be treated directly in the result section. The introduction could thus be largely shortened to highlight instead what general problems are being addressed, how they are addressed, and why these are significant.

I am grateful to the reviewer for this important suggestion. The Introduction summarized the results for the reviewers' convenience, but was not suitable for a broad readership. Now the Introduction has been streamlined to transport the reader to results as quickly as possible.

I hope that the new abstract and introduction make the "question" clearer: What is the function of this striking structure (Dm3 and TmY types and their wiring) in the fly optic lobe?

The meta-question is also made more explicit. Connectomics is uncovering an enormous amount of structural information, showering us with a huge number of "structures without functions." The paper is meant to explore the value of guessing the unknown functions by interpreting the connectome. This trend of starting from structure and predicting function is nontraditional but seems inescapable.

The discussion also feels like a continuation of the results section. It is presented as a list of independent and very specific sections, and maintain a speculative tone all along. These speculations will certainly interest specialists of the topic. However, it lacks a more general and comprehensive synthesis to highlight the significance of the neural motif identified to the general reader.

I have added a summary of predictions to the beginning of the Discussion.

We hope our review was helpful,

Yes it was!

Reviewer 3

Overview:

Seung provides a connectomic analysis for a set of connected neurons from the optic lobe of *Drosophila*, using connectomic electron microscope data from a single, female fly brain. Seung describes inputs and outputs of two major cell types, Dm3 and TmY, their subtypes, and the connectivity between them. Seung then uses the fact that their input neurons have an approximately one-to-one correspondence with single ommatidia in order to illustrate predictions for their functional receptive fields. Seung further describes downstream neurons, LC10e and LC15, that integrate inputs from these cell types. Again, Seung uses their inputs from Dm3 and TmY to make predictions about their functional receptive fields. The work is well written and easy to follow. Because there is little physiological data for these cell types, the connectomics approach taken here means the reader is left with several unanswered questions about the functional properties of these cells, and their role within the visual system. Nevertheless, this work provides a large volume of useful data in the process towards that goal. While the addition of functional analyses, whether from physiological recordings or from computational modelling,

would make this work an exceptional feat of progress, it would also make the manuscript enormous and more difficult to parse. As it stands, it provides helpful predictions that can be used as hypotheses for future studies.

Main concerns:

The main concern I have is with the use of the term "classical receptive field (CRF)". Although Seung does, at points, make it clear that this work cannot describe actual CRFs - which are determined by a neuron's functional response to the stimulus space - the term "CRF" is often used as though real CRFs are being reported in the results. I recommend using a variant of that term, e.g. pCRF (predicted CRF), or something similar, to avoid any confusion amongst less experienced readers who are less familiar with this nuance. I don't think it is necessary to use pCRF (or equivalent) for the phrase "beyond the CRF", where Seung describes modulatory effects on the CRF, whatever the actual CRF is.

Great point. I have decided to write "CRF" and "ERF" in quotes when a prediction is being made.

Minor concerns:

1) The Introduction is easy to follow, but reads more like a Discussion, laying out all the results that haven't yet been presented, and relating them to previously published work. I think the Introduction would greatly benefit from a description of more background literature that motivates the scientific work undertaken in this paper and the cells that have been chosen for analyses, before referring to the results.

The Abstract and Introduction have been completely rewritten to clarify and motivate. The Introduction is now concise, and takes the reader as quickly as possible to the results.

2) Introduction>par. 3:

The works of Hubel and Wiesel would be good to cite in this paragraph, as some of the earliest evidence for orientation selective receptive fields.

The citation to Hubel and Wiesel has now been moved close to the beginning of the paper.

3) Results>Dm3 receives input from collinear columns>par. 6:

Looks more like Dm3 pCRFs are 5x1, or even larger. Please describe in the text, or in the Methods, where the number 3 comes from.

I apologize for not making this explicit in the original manuscript. Dm3 and TmY "CRF"s were modeled as ellipses in Fig. 1g and Fig. 2k, in order to depict the variation across cells. In both figure panels, the scale bar indicates the hexagonal lattice spacing. The major radius σ_{major} of the ellipses is roughly 1.5 times the scale bar. So the length of the "CRF" is estimated to be $2\sigma_{\text{major}}$ which is roughly 3 lattice spacings. The calculation of the approximating ellipses is now explained in detail in the Methods. I've added a figure panel with boxplots of the major radii, so that the reviewer can see the variation in the major radius.

To supplement the ellipses, 1D projections of the predicted receptive fields have also been provided as further descriptive statistics about their sizes and shapes.

4) Results>Dm3 receptive field estimates>par. 4:

I find this paragraph difficult to determine which citation is responsible for which physiological property that is being reported. Are all relevant citations present here?

The main text now cites a review, and a new subsection in the Methods cites the primary sources for physiological properties of hexel cell types.

5) Results>Cross-orientation inhibition>par. 5:

Citation is needed for the sentence "These neurotransmitters are typically inhibitory in the fly brain, so Dm3 will be assumed inhibitory".

A subsection has been added to the Methods to give a full explanation of neurotransmitters and receptors, along with citations to the literature.

6) Results>LC10e pathways to the central complex:

This section appears to be purely discussion, rather than presenting any new results. Was there meant to be a reference to a figure here? If not, then I think it is best moved to the Discussion.

The finding that VES044 (now called LTe42) is a strong output of LC10e was new. But I have since realized that this cell type is just one of a longer list of targets of TmY and LC10e. As the current manuscript is already quite complicated, I have decided to defer analysis of downstream pathways to future work.

7) Results>ON-OFF input regions>par. 2 and par. 4:

Please show data for all reported results. Not only is this more transparent, but such data might also be useful for the reader.

"ON-OFF input regions" is no longer a separate section. Instead, these pathways are considered alongside other disynaptic pathways that are predicted to contribute to the ERF. They are mapped in the main Figures as well as in Data S3 and S4. The Reviewer may enjoy browsing through the latter, which gives some feeling for cell-to-cell variability.

8) Results>ON-OFF input regions>par. 4:

Is Fig. 6f mean to read 6b?

Yes I apologize for the erroneous cross reference.

9) Discussion:

I think it would be very helpful to list or tabulate the main predictions near the beginning of the Discussion, especially for those who have the capacity to verify the predictions in future work.

Great suggestion. Predictions are now summarized at the beginning of the Discussion.

10) Discussion>par.3>"are now empirical facts in the fly optic lobe":
I think stating this as fact is going too far, given that excitation and inhibition can only be verified by appropriate functional studies.

The reviewer recommends caution, so I have removed the claim. Also many caveats are included in the explanation of neurotransmitters and receptors in the Methods.

11) Discussion>OFF input regions as receptive field predictions>par. 1>"Orientation selective (OS) neurons were predicted to span a maximum of.....":

It seems as though the results presented in this study actually contradict the predictions of Horridge (2003), as they are all larger by at least one ommatidium. However, these predictions were made for the honeybee, so do not have to be supported accurately by the results here. It would be helpful to state this. Can the reasoning behind Horridge's predictions be applied to *Drosophila* to make new, *Drosophila*-specific predictions? That would be more informative than whether or not Horridge's predictions have been confirmed, per se, which doesn't further our scientific understanding of the results.

The reviewer is entirely correct that the behavioral experiments of Horridge need to be repeated for *Drosophila*, if a quantitative understanding is desired, because of interspecies differences. As now noted in the Discussion, for example, the interommatidial angle seems about twice as large for *Drosophila* as for the honeybee (Rigosi, Wiederman, and O'Carroll 2017).

And even if behavioral experiments were performed for *Drosophila*, we would still run into the challenge of defining Horridge's "maximum length of the orientation detector." Now that plausible neural candidates for the orientation detectors are known, it is unclear whether this length should be computed for Dm3 or TmY cells. And length can be quantified in multiple ways: I have chosen $2\sigma_{\text{major}}$, but $2.3\sigma_{\text{major}}$ is an alternate convention as it is the FWHM of a Gaussian with standard deviation σ_{major} .

Horridge's main point was actually more qualitative than quantitative: small vs. large. He argued that previous researchers had expected orientation detectors with large receptive fields, and accordingly employed visual stimuli that subtended large visual angles. But his behavioral experiments suggested that the orientation detectors must have much smaller receptive fields than expected. Confirmation of his qualitative point is provided by the present work.

12) Methods>Spatial coordinates in the optic lobe>par. 1>"These rules are simplified:...":

How is this simplification being used in the current study? As it is, this sentence confuses me, as the main text refers to the number of putative synapses as a measure of connectivity.

I apologize for being unclear. My point was that the current paper analyzes how connectivity depends on space, as well as how it depends on cell type. The paper on optic lobe cell types (Matsliah et al. 2023), in contrast, throws out space and characterizes only the dependence of connectivity on cell types.

13) Methods>Spatial coordinates in the optic lobe>par. 2>"I manually assigned all Mi1 cells to locations in a hexagonal lattice":
What considerations were made when doing this assignment manually? What caveats to this method are there, e.g. I suspect that this becomes a particularly unhelpful representation towards the periphery of the eye, or anywhere there is high curvature on the surface of the eye. Is this performed locally for each cell that was analysed, as I suspect a hexagonal lattice breaks down beyond a certain distance across the whole eye, e.g. because of dislocations in the lattice.

The columns at the extreme borders of the eye can be disorderly. I did not analyze defects in any systematic way, but my subjective impression is that there are few dislocations. The lattice of ommatidia in a wild type eye is known to contain very few dislocations (Kim et al. 2016). A detailed analysis of spatial layout is planned for a separate study, and is outside the scope of the present work.

14) Fig. 1 and other similar panels in all figures:
I strongly recommend adding axes to help the reader properly orient these diagrams.

3D crosshairs were in the original Figures, but I neglected to specify the directions in which they point. These are now specified in the Figure legends.

As a further aid to the reader, Data S1 now provides neuroglancer links, each of which makes an interactive visualization available via a single click. Neuroglancer links are also embedded in the Figure legends.

15) Figs. 1, 2, 5 and 6:
There are some long numbers in the captions that don't seem to represent anything, presumably a formatting error.

The long numbers were Cell IDs. These have now been replaced by the neuroglancer links in the Figure legends and Data S1.

16) Fig. S2:
It looks as though, when grouped in panel D, TmY4/TmY9 receive more Tm1 inputs than the sum of TmY9qL and TmY9q in panel B, with TmY4 not even showing on panel B.

Thanks for pointing this out. I finally realized that the plots were misleading because the x axis did not start at zero! In the revised figure, I reset the x limits, and now panels B and D look consistent. I also increased the number of cell types on the vertical axis of panel B, so that TmY4 shows up.

References

- Ammer, Georg, Etienne Serbe-Kamp, Alex S. Mauss, Florian G. Richter, Sandra Fendl, and Alexander Borst. 2023. "Multilevel Visual Motion Opponency in *Drosophila*." *Nature Neuroscience* 26 (11): 1894–1905.
- Arenz, Alexander, Michael S. Drews, Florian G. Richter, Georg Ammer, and Alexander Borst. 2017. "The Temporal Tuning of the *Drosophila* Motion Detectors Is Determined by the Dynamics of Their Input Elements." *Current Biology: CB* 27 (7): 929–44.
- Borst, Alexander, and Moritz Helmstaedter. 2015. "Common Circuit Design in Fly and Mammalian Motion Vision." *Nature Neuroscience* 18 (8): 1067–76.
- Clark, Damon A., and Jonathan B. Demb. 2016. "Parallel Computations in Insect and Mammalian Visual Motion Processing." *Current Biology: CB* 26 (20): R1062–72.
- Davis, Fred P., Aljoscha Nern, Serge Picard, Michael B. Reiser, Gerald M. Rubin, Sean R. Eddy, and Gilbert L. Henry. 2020. "A Genetic, Genomic, and Computational Resource for Exploring Neural Circuit Function." *eLife* 9 (January). <https://doi.org/10.7554/eLife.50901>.
- Dorkenwald, Sven, Arie Matsliah, Amy R. Sterling, Philipp Schlegel, Szi-Chieh Yu, Claire E. McKellar, Albert Lin, et al. 2023. "Neuronal Wiring Diagram of an Adult Brain." *bioRxiv : The Preprint Server for Biology*, July. <https://doi.org/10.1101/2023.06.27.546656>.
- Drews, Michael S., Aljoscha Leonhardt, Nadezhda Pirogova, Florian G. Richter, Anna Schuetzenberger, Lukas Braun, Etienne Serbe, and Alexander Borst. 2020. "Dynamic Signal Compression for Robust Motion Vision in Flies." *Current Biology: CB* 30 (2): 209–21.e8.
- Eckstein, Nils, Alexander S. Bates, M. Du, V. Hartenstein, Gsxe Jefferis, and J. Funke. 2020. "Neurotransmitter Classification from Electron Microscopy Images at Synaptic Sites in *Drosophila*." *BioRxiv*. <https://www.biorxiv.org/content/10.1101/2020.06.12.148775v3.abstract>.
- Eschbach, Claire, Akira Fushiki, Michael Winding, Casey M. Schneider-Mizell, Mei Shao, Rebecca Arruda, Katharina Eichler, et al. 2020. "Recurrent Architecture for Adaptive Regulation of Learning in the Insect Brain." *Nature Neuroscience* 23 (4): 544–55.
- Gatto, Elia, Olli J. Loukola, Maria Elena Miletto Petrazzini, Christian Agrillo, and Simone Cutini. 2022. "Illusional Perspective across Humans and Bees." *Vision (Basel, Switzerland)* 6 (2). <https://doi.org/10.3390/vision6020028>.
- Horridge, Adrian. 2009. *What Does the Honeybee See? And How Do We Know?: A Critique of Scientific Reason*. ANU Press.
- Hulse, Brad K., Hannah Haberkern, Romain Franconville, Daniel Turner-Evans, Shin-Ya Takemura, Tanya Wolff, Marcella Noorman, et al. 2021. "A Connectome of the *Drosophila* Central Complex Reveals Network Motifs Suitable for Flexible Navigation and Context-Dependent Action Selection." *eLife* 10 (October). <https://doi.org/10.7554/eLife.66039>.
- Keleş, Mehmet F., Ben J. Hardcastle, Carola Städele, Qi Xiao, and Mark A. Frye. 2020. "Inhibitory Interactions and Columnar Inputs to an Object Motion Detector in *Drosophila*." *Cell Reports* 30 (7): 2115–24.e5.
- Kim, Sangwoo, Justin J. Cassidy, Boyuan Yang, Richard W. Carthew, and Sascha Hilgenfeldt. 2016. "Hexagonal Patterning of the Insect Compound Eye: Facet Area Variation, Defects, and Disorder." *Biophysical Journal* 111 (12): 2735–46.
- Kurmangaliyev, Yerbol Z., Juyoun Yoo, Javier Valdes-Aleman, Piero Sanfilippo, and S. Lawrence Zipursky. 2020. "Transcriptional Programs of Circuit Assembly in the *Drosophila* Visual System." *Neuron* 108 (6): 1045–57.e6.
- Liu, Wendy W., and Rachel I. Wilson. 2013. "Glutamate Is an Inhibitory Neurotransmitter in the *Drosophila* Olfactory System." *Proceedings of the National Academy of Sciences of the United States of America* 110 (25): 10294–99.
- Matsliah, Arie, Szi-Chieh Yu, Krzysztof Kruk, Doug Bland, Austin Burke, Jay Gager, James Hebditch, et al. 2023. "Neuronal 'Parts List' and Wiring Diagram for a Visual System." *bioRxiv : The Preprint Server for Biology*, October. <https://doi.org/10.1101/2023.10.12.562119>.
- Rigosi, Elisa, Steven D. Wiederman, and David C. O'Carroll. 2017. "Visual Acuity of the Honey Bee Retina and the Limits for Feature Detection." *Scientific Reports* 7 (April): 45972.

- Sanes, Joshua R., and S. Lawrence Zipursky. 2010. "Design Principles of Insect and Vertebrate Visual Systems." *Neuron* 66 (1): 15–36.
- Schlegel, Philipp, Yijie Yin, Alexander Shakeel Bates, Sven Dorkenwald, Katharina Eichler, Paul Brooks, Daniel S. Han, et al. 2023. "Whole-Brain Annotation and Multi-Connectome Cell Typing Quantifies Circuit Stereotypy in *Drosophila*." *bioRxiv : The Preprint Server for Biology*, July. <https://doi.org/10.1101/2023.06.27.546055>.
- Städele, Carola, Mehmet F. Keleş, Jean-Michel Mongeau, and Mark A. Frye. 2020. "Non-Canonical Receptive Field Properties and Neuromodulation of Feature-Detecting Neurons in Flies." *Current Biology: CB* 30 (13): 2508–19.e6.
- Zhao, Arthur, Eyal Gruntman, Aljoscha Nern, Nirmala A. Iyer, Edward M. Rogers, Sanna Koskela, Igor Siwanowicz, et al. 2022. "Eye Structure Shapes Neuron Function in *Drosophila* Motion Vision." *bioRxiv*. <https://doi.org/10.1101/2022.12.14.520178>.

Reviewer Reports on the First Revision:

Referees' comments:

Referee #1 (Remarks to the Author):

I am happy with the extensive and substantive revisions that the author has made. I think the new figures and text have improved the manuscript greatly!

Referee #2 (Remarks to the Author):

The author made significant changes and efforts to answer our comments. The overall re-organisation makes the manuscript a much more digestible read and the predictions (now highlighted in the subheadings) perfectly highlights the contribution of this work. The representation of variations in receptive fields through population of ellipses (and the associated quantification through boxplots) is a wonderful idea. Overall, most of our original concerns have been lifted. I remain however unconvinced by the answers given to our criticisms about the originality and significance of the work, to which I respond below.

Author : It is difficult or impossible to discover neural mechanisms based on behavioral experiments alone. If it were so easy, there would be no need for neuroscience.

This tautological argument confounds the level of computation (inferable from behaviour) and their neural implementation; and feels ironical given that the neural mechanisms put forward in the current work have actually been specifically suggested by behavioural experiments alone (e.g., Srinivasan et al., 1994 : extraction of edges along three directions following the lattice; Horridge: edge detector extending over 3 ommatidia lengths, as acknowledged in the discussion).

Neuroscience is obviously required to demonstrate any neural implementation. However, David Marr (and others later) have well explained why the prior knowledge given by behavioural experiments is fundamental to guide neurosciences . The current work does not deviate from this rule, in contrast, it provides a beautiful example of it.

Krakauer, John W., et al. "Neuroscience needs behavior: correcting a reductionist bias." *Neuron* 93.3 (2017): 480-490.

(I do not imply that this should be cited in the manuscript)

Author : Regarding novelty, many people have been surprised that structural analysis can make such detailed predictions about visual responses of a neural circuit, when none of the neurons in that circuit have ever been studied by physiologists.

Author : [...] historically the discovery of compass neurons was made by physiologists, and later on connectomic analyses attempted to explain the mechanisms underlying what the physiologists had discovered. One can imagine a counterfactual history in which (Hulse et al. 2021) predicted compass neurons by analyzing the central complex wiring diagram, and physiologists later confirmed the prediction. But that's not how history unfolded.

Similarly, physiologists already knew that DANs carry reward/punishment/teaching signals, and MBONs signal the learned valences of stimuli. The connectome helps one figure out the mechanisms supporting those functions (Eschbach et al. 2020). The novelty of the present work is that I am predicting the functions of all cell types in a novel circuit BEFORE the physiologists have done their experiments. This step might seem obvious and natural to the reviewer, but everyone I've met seems surprised by it.

I agree that the current work pushes forward the existence of functions that have not been clearly demonstrated by physiologists or (I now acknowledge regarding the illusory contours) behavioural studies in insects, but the ideas of these functions, as well as the way they can be neurally implemented, are not novel, and this, according to me, does not make the current approach fundamentally different from the other connectomic work cited above. What comes BEFORE, both here and in the work cited, is an a priori clear notion of the expected types of computation to be found in this part of the brain (and I think this is a correct way to do it indeed). A difference I see is that the Mushroom body or central complex are central brain areas, and therefore early physiology (or lesion) studies were required to target what type of functions they may implement at the first place (among the function expected from behavioural studies). Here the work is achieved in a sensory lobe, nowadays well-studied, and providing all the necessary a priori knowledge that the (previously suggested) type of visual computation highlighted here might be found there rather than somewhere else.

Note that the works based on the connectomic of the Mushroom body and central complex (such as the ones mentioned above) did not merely confirm what physiologists said (as implied in the author response), but also reported unexpected motifs and speculated about their functions (using prior knowledge ultimately stemming from behaviour, such as, for instance, the notions of predictive coding (Mushroom body), or the need to compare between a current and goal heading, (central complex)), as achieved here. I am therefore, indeed, not surprised that one can derive functional prediction through connectomic before physiology (but still, after behaviour).

This said, I absolutely believe that deciphering neural networks through connectomics is an essential step to understand neural implementation fully, and the current work forms a particularly remarkable contribution of this kind.

Author : The present work suggests a new approach to studying illusory contours starting with neurophysiology. First the CRFs and ERFs of Dm3 and TmY cells should be measured and compared with my predictions. The findings should be used to constrain a neural network model based on the connectivity patterns described by this paper. Then one should search for a visual stimulus that excites a TmY cell in the model even without stimulus contrast inside the TmY CRF. Then this visual stimulus should be applied to a real fly, and TmY responses should be measured. If this stimulus results in neurophysiological evidence of illusory contour responses, then one can proceed to use the same stimulus in behavioral experiments.

Yes, but this approach is not new, it corresponds to the natural second step (back to behaviour) needed to confirm a suggested neural implementation, as advocated by Marr (see figure 2 in Krakauer et al., 2017 mentioned in my first comment above). I do believe that the clear and targeted predictions laid out by the current work will solidly remotivate such a second step regarding the detection of illusory contours, this time in drosophila.

Author : Parallels have generally been drawn with vertebrate retina, and this reference is no exception. The present work is novel because it points out possible parallels with mammalian cortex.

This parallel was also put forward and was the main point of Srinivasan et al., 1993 work published in nature 20 years ago.

Srinivasan, et al. (1993). Is pattern vision in insects mediated by 'cortical' processing?. Nature, 362(6420), 539-540.

Overall, the work is remarkable for its quality and quantity, and, even if it does not bring novel concepts regarding visual neural mechanisms, it will surely motivate future research regarding insect visual perception through the precise predictions given for drosophila, as well as serve as an example of the powers of analysing connectomic data.

Referee #3 (Remarks to the Author):

I'd like to thank the author for their hard work in addressing the issues I had raised. The author has provided a satisfactory response to all of my comments. I have no further issues to raise with the current version of the manuscript.

The work provides a helpful step in progressing research into *Drosophila* vision. The predictions provided by this manuscript are concrete and appear to be verifiable, if the specific neurons about which predictions are made can be targeted experimentally.

Author Rebuttals to First Revision:

Referee #2 acknowledged that the paper “will surely motivate future research regarding insect visual perception through the precise predictions given for drosophila,” and Referee #3 was happy that “The predictions provided by this manuscript are concrete and appear to be verifiable.” The referees are correct in pointing out that the novelty is the precision and concreteness of the predictions. Therefore I have created a penultimate section “Predictions and their limitations.” This section not only details the predictions, but also describes what we would learn if the predictions turn out to be incorrect. I also explain how the relation between modeling and physiology is transformed by interpreting the wiring diagram. The final Discussion section zooms out to explain the context of artificial intelligence, mammalian V1, and insect visual behaviors, as well as suggest questions for research on neural development.